# Cross-talks of glycosylphosphatidylinositol biosynthesis with glycosphingolipid biosynthesis and ER-associated degradation

Yicheng Wang [1,2], Yusuke Maeda [1], Yi-Shi Liu[3], Yoko Takada [2], Akinori Ninomiya [1], Tetsuya Hirata[1,2,4], Morihisa Fujita[3], Yoshiko Murakami [1,2] & Taroh Kinoshita [1,2]✉

Glycosylphosphatidylinositol (GPI)-anchored proteins and glycosphingolipids interact with each other in the mammalian plasma membranes, forming dynamic microdomains. How their interaction starts in the cells has been unclear. Here, based on a genome-wide CRISPR-Cas9 genetic screen for genes required for GPI side-chain modification by galactose in the Golgi apparatus, we report that β1,3-galactosyltransferase 4 (B3GALT4), the previously characterized GM1 ganglioside synthase, additionally functions in transferring galactose to the *N*-acetylgalactosamine side-chain of GPI. Furthermore, B3GALT4 requires lactosylceramide for the efficient GPI side-chain galactosylation. Thus, our work demonstrates previously unexpected functional relationships between GPI-anchored proteins and glycosphingolipids in the Golgi. Through the same screening, we also show that GPI biosynthesis in the endoplasmic reticulum (ER) is severely suppressed by ER-associated degradation to prevent GPI accumulation when the transfer of synthesized GPI to proteins is defective. Our data demonstrates cross-talks of GPI biosynthesis with glycosphingolipid biosynthesis and the ER quality control system.

[1] Research Institute for Microbial Diseases, Osaka University, Suita, Osaka 565-0871, Japan. [2] WPI Immunology Frontier Research Center, Osaka University, Suita, Osaka 565-0871, Japan. [3] Key Laboratory of Carbohydrate Chemistry and Biotechnology, Ministry of Education, School of Biotechnology, Jiangnan University, Wuxi, Jiangsu 214122, China. [4] Present address: Center for Highly Advanced Integration of Nano and Life Sciences (G-CHAIN), Gifu University, 1-1 Yanagido, Gifu-City Gifu 501-1193, Japan ✉email: tkinoshi@biken.osaka-u.ac.jp

Glycosylphosphatidylinositol (GPI) is a complex glycolipid for post-translational modification of many cell-surface proteins in eukaryotic cells[1]. To date, more than 150 human proteins have been confirmed as GPI-anchored proteins (GPI-APs)[2]. The structure of the core backbone of GPI, which is conserved in eukaryotic organisms, is EtNP-6Manα−2Manα −6Manα−4GlcNα−6Inositol-phospholipid (where EtNP, Man and GlcN are ethanolamine phosphate, mannose and glucosamine, respectively) (Fig. 1a). GPI is synthesized in the endoplasmic reticulum (ER) followed by transfer of GPI to proteins that have a C-terminal GPI-attachment signal peptide. The GPI-attachment signal peptide is removed and replaced with GPI by the GPI-transamidase (GPI-Tase) complex to form immature GPI-APs. Nascent GPI-APs undergo structural remodeling in the ER and the Golgi apparatus. The inositol-linked acyl chain is deacylated and the EtNP side branch is removed from the second Man (Man2) for efficient ER to Golgi transport. In the Golgi, GPI fatty acid remodeling occurs, in which an *sn*-2-linked unsaturated fatty acyl chain is removed and reacylated with a saturated chain,

usually stearic acid. Fatty acid remodeling is crucial for lipid-raft association of GPI, a feature of GPI-APs.

The structural variation of GPI anchors is introduced by side-chain modifications[1]. Structural studies of GPIs from some mammalian GPI-APs indicated that the first mannose (Man1) is often modified with β1,4-linked *N*-acetylgalactosamine (GalNAc)[3,4]. This modification is mediated by PGAP4 (Post-GPI attachment to proteins 4 also known as TMEM246), a recently identified Golgi-resident, GPI-specific GalNAc-transferase[5]. The GalNAc side-chain can be further modified with β1,3 galactose (Gal) by an unknown galactosyltransferase (Gal-T) and then with sialic acid (Sia) (Fig. 1a)[4]. Some GPI-APs have forth Man (Fig. 1a), which is transferred by an ER-resident mannosyltransferase PIGZ to the third Man before transfer of the GPI to proteins. These side-chains of GPI do not seem to affect cell surface expression of GPI-APs in cultured cells, but might affect the properties of some GPI-APs. An example is the role of a sialylated GPI side-chain in prion protein: It was shown by in vitro studies that lack of Sia in the GPI side-chain of PrP^c slowed generation of pathogenic PrP^sc after infection of

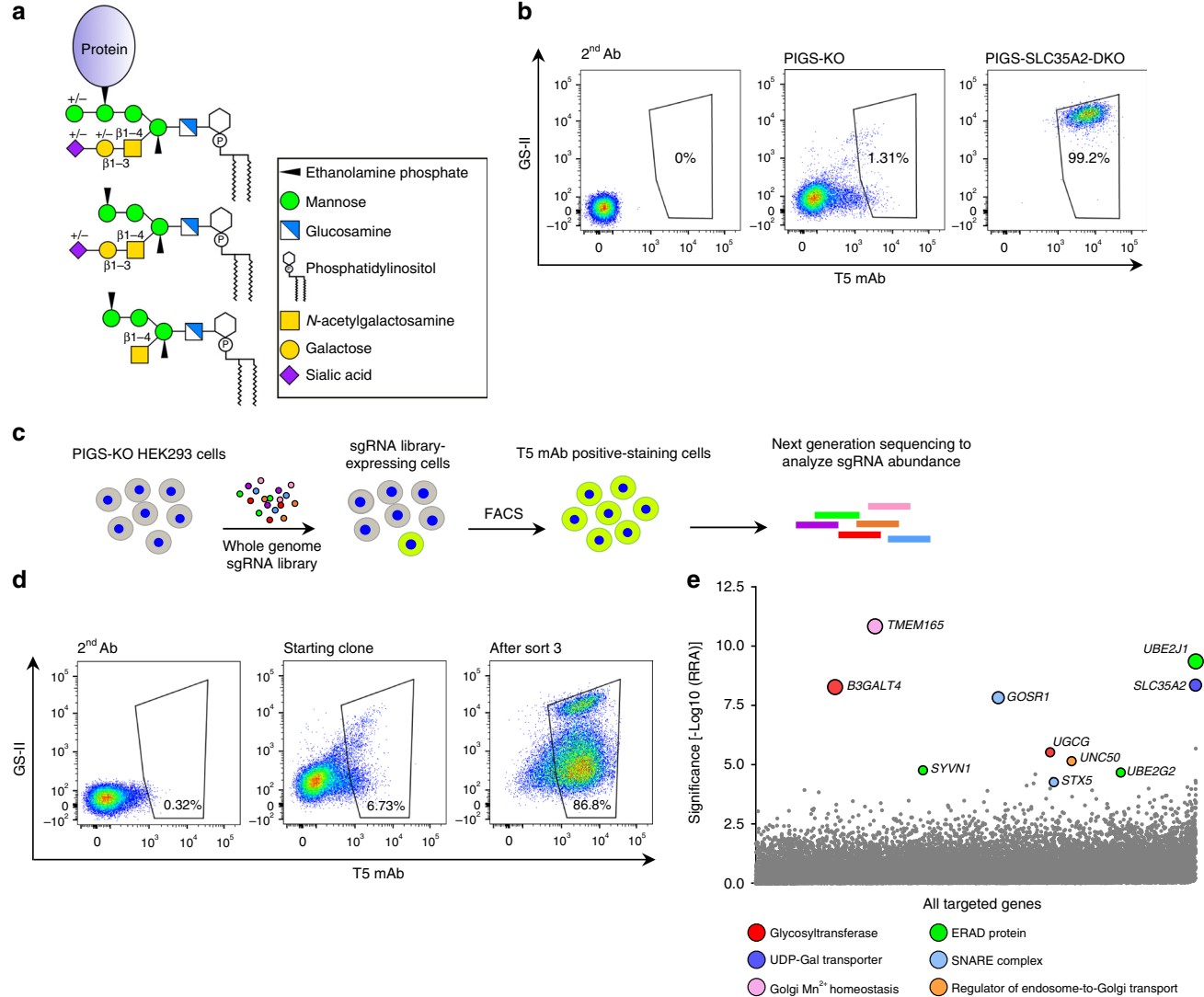

**Fig. 1 CRISPR-based genetic screen identified genes involved in galactosylation of GPI. a** GPI-APs and free GPIs with side-chain modifications. Monosaccharide symbols are drawn according to the symbol nomenclature for glycans[77]. **b** Flow cytometry analysis of PIGS knockout (PIGS-KO) and PIGS-SLC35A2-DKO HEK293 cells. Cells were double stained with T5 mAb for free GPI-GalNAc and GS-II lectin. **c** Schematic depicting a FACS-based genome-wide CRISPR screen for GPI-galactosylation using PIGS-KO HEK293 cells. **d** Flow cytometry analysis of starting clone and sort3 of PIGS-KO cells stained with T5 mAb and GS-II lectin. **e** Gene scores in unsorted versus sort3 PIGS-KO cells. Ten top-ranking ER or Golgi protein-coding genes are shown. Bubble size shows the number of active sgRNAs per gene. See also Supplementary Fig. 1e, f.

cultured cells, suggesting involvement of the sialylated GPI side-chain in conversion of PrP[c] to PrP[sc][6]. A recent in vivo study showed that the presence of PrP[c], which lacks Sia in the GPI, was protective against infection by PrP[sc][7]. However, another report suggested that GPI sialylation of PrP[c] is regulated in a host-, cell- or tissue-specific manner and that PrP[c] proteins with both sialo- and asialo-GPI were converted to PrP[sc][8]. Because current understanding of GPI sialylation is still incomplete, identification of the relevant GPI-Gal-T would be helpful for further understanding of the GPI-GalNAc side-chain modification pathway, which is required for understanding of the significance of GPI structural variation in vivo.

Glycosphingolipids (GSLs) are glycolipids consisting of ceramide and an oligosaccharide chain[9]. GPI-APs were proposed to associate with lipid rafts, microdomains enriched in cholesterol and sphingolipids, from the trans-Golgi network (TGN) to the plasma membrane[10,11]. Evidence suggests that lipid rafts are highly dynamic and heterogeneous[12], and GPI-APs are thought to be organized in cholesterol- and sphingomyelin-dependent submicron-sized domains in live cell membranes[13]. The current view of the functional microdomains is that they might be present in the Golgi[14,15]. Interactions between GPI-APs and GSLs are rather transient[16]. Although both GPIs and GSLs are synthesized in the ER and the Golgi, evidence for their functional interactions in the Golgi has not been reported, and whether GPI-APs associate with specific GSLs in the Golgi remains unclear.

Here, we report genome-scale CRISPR-Cas9 knockout (GeCKO) screening for genes involved in galactosylation of the GalNAc side-chain of GPI, particularly the GPI-Gal-T. Through the screening, we define key enzymes and regulators of the GPI galactosylation pathway. Of particular interest, β1,3-galactosyltransferase 4 (B3GALT4, also known as GM1 synthase), a Gal-T thought to be limited to the GSL pathway, is the only candidate for GPI-Gal-T identified by our GeCKO screen. B3GALT4 transfers Gal to a β1,4-linked GalNAc side-chain of GPI. We also demonstrate the requirement for lactosylceramide (LacCer) for efficient galactosylation of GPI-GalNAc. In addition, we identify components of an ER-associated degradation (ERAD) pathway through the same screening: we show that the ERAD is not involved in regulation of GPI-GalNAc galactosylation but instead negatively regulates GPI biosynthesis when too much GPI is accumulated.

## Results

**Identification of genes involved in galactosylation of GPI.** To identify the GPI-Gal-T that transfers Gal to GPI-GalNAc, we established a forward genetic screening system. For enrichment of mutant cells defective in galactosylation of GPI-GalNAc, a probe that detects the galactosylation status of GPI-GalNAc is required. T5-4E10 monoclonal antibody (T5 mAb) recognizes non-protein-linked, free GPI when it has the GalNAc side-chain but only when the GalNAc is at the non-reducing end (Fig. 1a, bottom), i.e., T5 mAb does not bind free GPI when GalNAc is galactosylated (Fig. 1a, middle)[5,17,18]. Therefore, T5 mAb is useful to identify mutant cells defective in galactosylation of GPI-GalNAc. As a parental cell that receives a genome-wide guide RNA library, we used PIGS-knockout (KO) HEK293 cells[19]. PIGS is one of the subunits of GPI-Tase and essential for transfer of GPI to proteins; therefore, all GPIs become free, non-protein-linked GPI in PIGS-KO cells. PIGS-KO HEK293 cells were only barely stained by T5 mAb, whereas PIGS and SLC35A2 double knockout (DKO) HEK293 cells, in which UDP-Gal is not available for Gal-Ts, were strongly stained by T5 mAb (Fig. 1b). This result indicates that almost all GPI-GalNAc was galactosylated in PIGS-KO HEK293

cells (shown schematically in Supplementary Fig. 1a). The PIGS-SLC35A2 DKO cells were also strongly stained by *Griffonia simplicifolia* lectin II (GS-II) because of exposure of *N*-acetylglucosamine (GlcNAc) on *N*-glycans and other oligosaccharides by defects in galactosylation (Fig. 1b, right, and Supplementary Fig. 1a). These results indicate that if GPI-Gal-T is lost, PIGS-KO HEK293 cells would be positively stained by T5 mAb but not by GS-II. In contrast, if the general galactosylation ability is lost, the PIGS-KO cells would be positively stained by both T5 mAb and GS-II.

Based on this principle, we transduced PIGS-KO HEK293 cells with a lentiviral short guide RNAs (sgRNA) library (GeCKO v2)[20]. We then cultured the cells for 2 weeks, and cells positively stained by T5 mAb were enriched by cell sorting (Fig. 1c). After three rounds of cell sorting and culture (sort3 cells), most cells became positively stained by T5 mAb, and some were also positively stained by GS-II lectin (Fig. 1d). The sgRNA abundance in the sorted and unsorted cells was determined by deep sequencing and analyzed by the Model-based Analysis of Genome-wide CRISPR-Cas9 Knockout (MAGeCK) method[21]. We calculated significance scores based on the robust ranking aggregation (RRA) algorithm as the fold-change in the abundance of multiple sgRNAs targeting the same gene. We observed hits that were highly enriched for some genes in sort3 cells, including the expected *SLC35A2* (Fig. 1e), and a significant enrichment of individual sgRNAs for those genes (Supplementary Fig. 1e, f).

**Validation and phenotypic grouping of CRISPR screen hits.** To validate roles of candidate genes in galactosylation of GPI-Gal-NAc, we knocked out each of the 10 top-ranking hits from PIGS-KO HEK293 cells using the CRISPR-Cas9 system. The KO cells were analyzed by staining with three probes: T5 mAb to determine the galactosylation status of GPI-GalNAc; cholera toxin B-subunit (CTxB) to determine GM1 levels because GSL biosynthetic genes were among the candidates; and GS-II lectin to determine any effect on *N*-glycans and other oligosaccharides (Fig. 2a). Consistent with the results of the genetic screening (Fig. 1e), all these KO cells showed positive T5 mAb staining (Fig. 2b, c), indicating that all the tested top candidate genes are real targets.

These genes were classified into four groups based on the mean fluorescence intensity (MFI) of staining by these three probes (Fig. 2d). The first group consisted of *SLC35A2* and *TMEM165*, whose defects caused loss of GM1 and affected *N*-glycans. TMEM165 is involved in Golgi $Mn^{2+}$ homeostasis for proper glycosylation of *N*-glycans, *O*-glycans and glycolipids[22], suggesting a requirement for $Mn^{2+}$ for Gal modification of GPI-GalNAc. The second group consisted of *GOSR1* (encoding GS28), *STX5* and *UNC50*. GS28 and STX5 are components of the Golgi target-membrane-bound (t)-SNARE-complex[23], and UNC50 is a factor involved in intra-cellular trafficking[24]. Their defects induced mildly positive T5 mAb staining, decreased CTxB staining, and slightly increased levels of staining by GS-II lectin. It seems likely that intracellular trafficking of GPI-Gal-T itself and/or other glycosyltransferases (GTs) was affected in this group, causing the changes in the profiles of all three probes.

The third group included two GTs *B3GALT4* and *UGCG*, whose defects caused loss of GM1 but did not result in positive staining by GS-II. B3GALT4 has been thought to be a GSL specific Gal-T[25,26], while UGCG, a ceramide glucosyltransferase, catalyzes the first glycosylation in GSL synthesis[27]. We chose these GTs for further analysis.

The fourth group consisted of *SYVN1* (encoding HRD1), *UBE2J1* and *UBE2G2*, key members of the ERAD pathway[28,29]. Whereas ERAD was implicated in degradation of misfolded GPI-

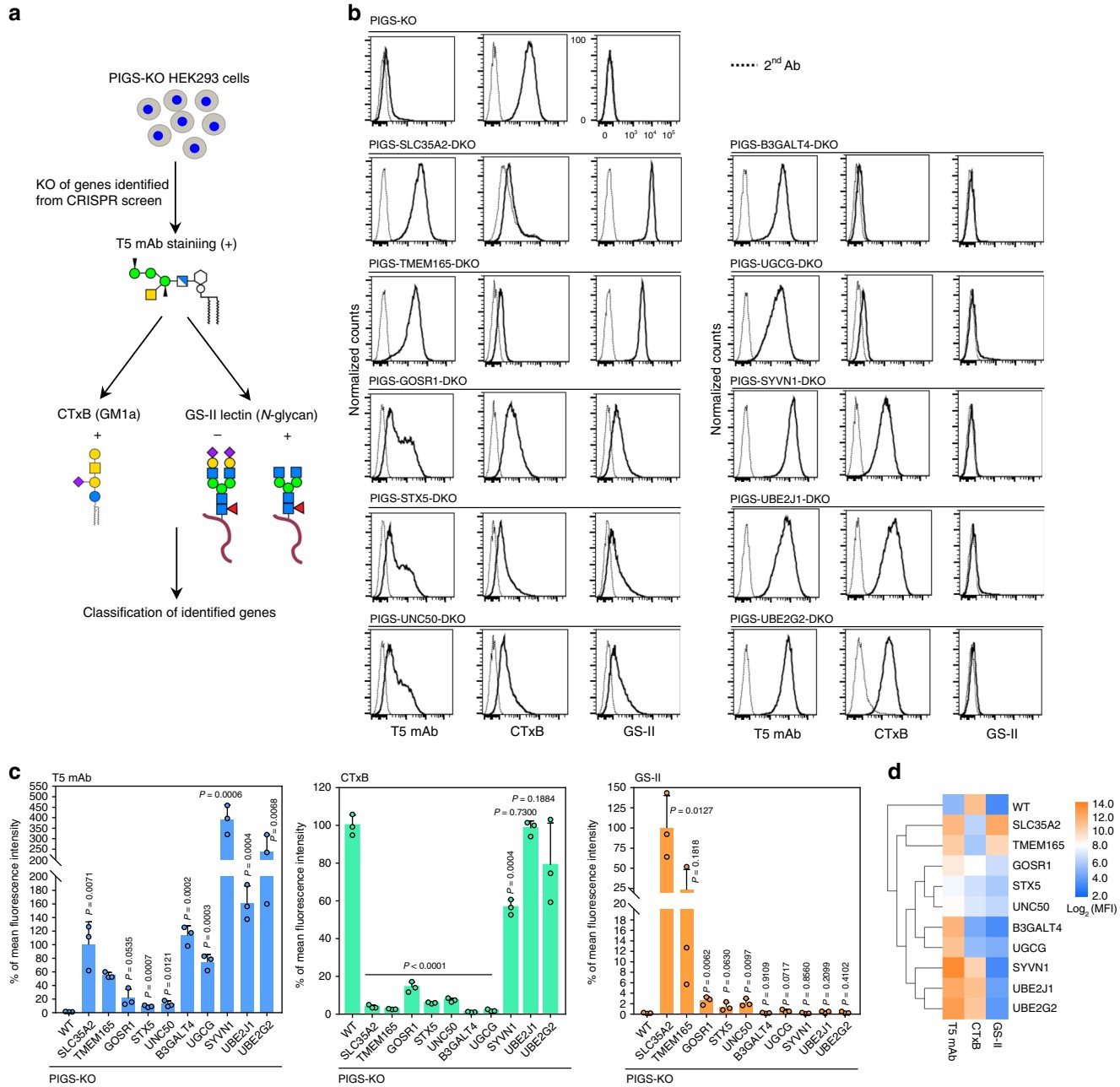

**Fig. 2 Validation and phenotypic grouping of CRISPR screen hits. a** Schematic depicting validation of the CRISPR screen by further KO of individual genes and stained by T5 mAb, Cholera enterotoxin subunit B (CTxB), and GS-II lectin. **b** Flow cytometry analysis of PIGS-KO HEK293 cells defective in each gene among 10 top-ranking protein-coding genes identified by our CRISPR screen. Cells were stained with T5 mAb, GS-II lectin and CTxB, respectively. See also Supplementary Fig. 1g–k and Supplementary Data 2 and 3. **c** Quantitative data of mean fluorescence intensity (MFI) from three independent experiments (mean ± SD, $n = 3$) based on stainings of T5 mAb (left), CTxB (middle), and GS-II (right). $P$ values are from $t$ test (unpaired and two-tailed) with comparisons to control (PIGS-KO). **d** Hierarchical clustering of glycan profiles analyzed by flow cytometry analyses, showing the effect (log$_2$ normalized MFI values) by each gene knockout based on staining of three probes. Source data are provided as a Source Data file.

APs[30,31], its relationship to GPI has not been reported. Knockout of *SYVN1*, *UBE2J1* and *UBE2G2* greatly increased T5 mAb staining without affecting CTxB and GS-II staining profiles. Because of this selective effect on GPI, we chose these ERAD components for further study.

Overall, this phenotypic classification clearly grouped the top hits from the screen and helped identify target genes for further studies.

**B3GALT4 transfers Gal to both GM2 and GPI-GalNAc.** We first focused on B3GALT4, which was thought to be a gangliosides-

specific Gal-T. B3GALT4 transfers a β1,3 Gal from UDP-Gal to GalNAc(β1-4)Gal(β1-4)-R of GA2, GM2, GD2, and GT2 to form GA1 (asialo-GM1a), GM1a, GD1b, and GT1c, respectively[25,26]. Given the structural similarity between GPI-GalNAc and these known acceptor substrates of B3GALT4 (Fig. 3a), we hypothesized that B3GALT4 also galactosylates GPI-GalNAc. Flow cytometric analysis showed that knockout of B3GALT4 from PIGS-KO HEK293 cells greatly increased cell surface T5 mAb staining and abolished CTxB binding (Fig. 3b, compare top and middle), and these phenotypes were normalized by transfection of *B3GALT4* cDNA (bottom). Immunofluorescence staining of such cells

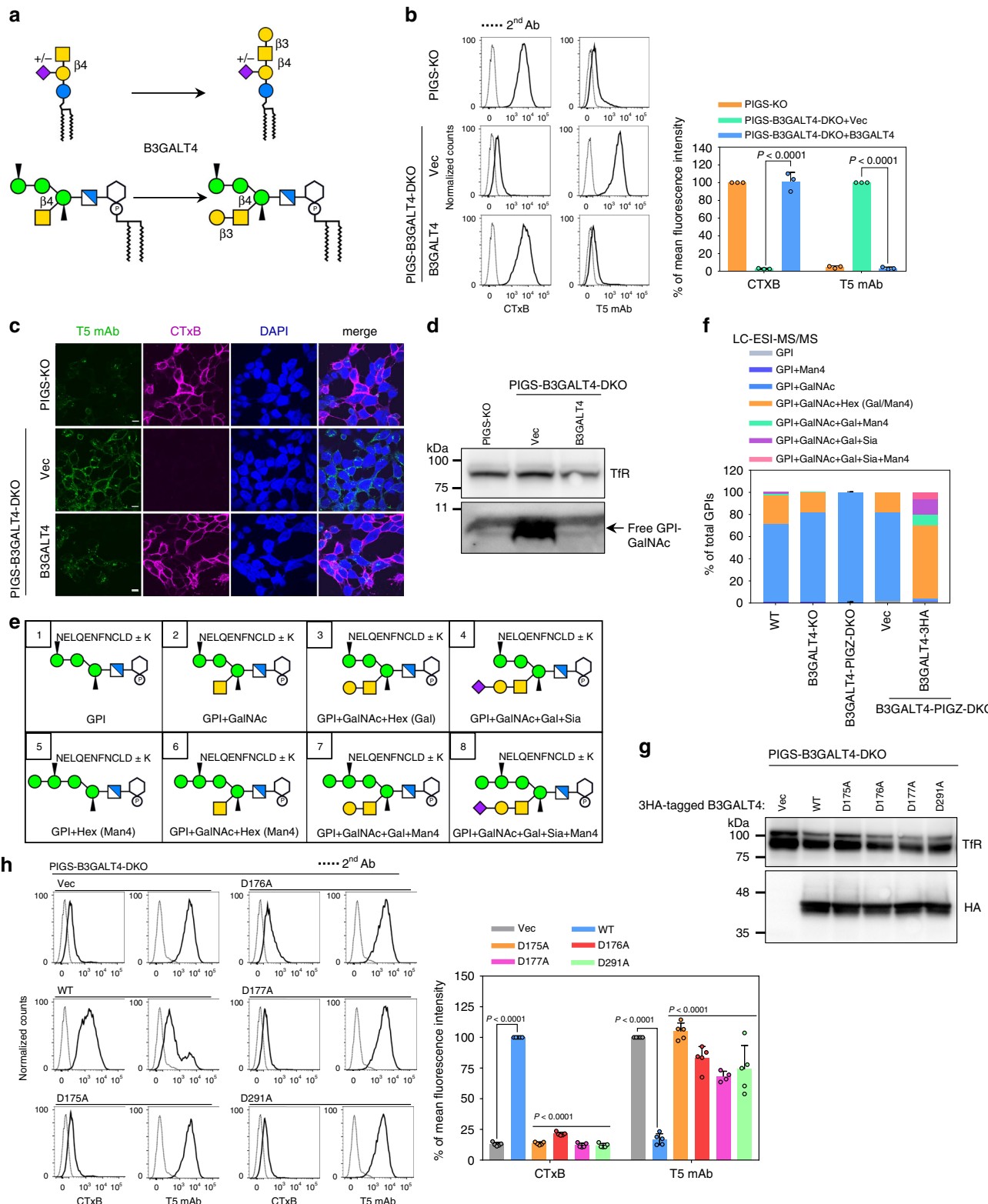

confirmed similar results, demonstrating that both GM1 and free GPI-GalNAc were mostly detected on the cell surfaces (Fig. 3c). Western blotting with T5 mAb revealed the presence of free GPI-bearing only GalNAc below the 11-kDa marker when *B3GALT4* was knocked out from PIGS-KO HEK293 cells (Fig. 3d, middle lane). The band disappeared after transfection of FLAG-tagged *B3GALT4* cDNA, indicating recovery of galactosylation of free GPI-GalNAc (right lane) (see Supplementary Fig. 2a for

expression of 3FLAG-B3GALT4). Taken together, these results verified that B3GALT4 is essential for Gal modification of the GalNAc side-chain of free GPI.

To determine whether B3GALT4 also transfers Gal to the GalNAc side-chain of protein-linked GPI-anchor, we used liquid chromatography-electrospray ionization-tandem mass spectrometry to identify GPI glycoforms of GPI-APs. Based on the previous observation of Gal modification of GPI-GalNAc in

**Fig. 3 B3GALT4 transfers Gal to both GM2 and GPI-GalNAc. a** B3GALT4, which is required for transfer Gal to GA2 and GM2 to generate GA1 and GM1a, is the candidate for GPI-Gal-T. **b** Left: PIGS-KO (top) and PIGS-B3GALT4-DKO HEK293 cells stably expressing pME-Puro (Vec) or pME-Puro-hB3GALT4-3FLAG (B3GALT4) were stained with CTxB and T5 mAb. Right: Quantitative data of MFI from three independent experiments (mean ± SD, $n = 3$). $P$ values are from $t$ test (unpaired and two-tailed) with comparisons to vector control. See also Supplementary Fig. 2a. **c** Representative images of cells labeled with CTxB on ice, fixed, and stained with T5 mAb. Scale bar, 10 μm. **d** Western blotting of free GPI-GalNAc. Lysates of PIGS-KO and PIGS-B3GALT4-DKO HEK293 cells stably expressing Vec or B3GALT4 were analyzed by Western blotting. TfR, a loading control. **e** Expected structures of GPI-bearing C-terminal peptides from HFGF-CD59 that was released from HEK293 cells by PI-PLC and digested with trypsin. The C-terminal peptides linked to GPI of glycoforms 1–8 are shown. **f** Quantitative data of LC-ESI-MS/MS analysis. Percentage of total intensity was calculated from the peak areas. See also Supplementary Fig. 2b–d and 3, and Supplementary Table 2. **g** Expression of catalytic mutant of B3GALT4 confirmed by western blotting. TfR, a loading control. **h** Left: Flow cytometry analyses of PIGS-B3GALT4-DKO cells transiently expressing pME-hB3GALT4-3HA (WT), -hB3GALT4-D175A-3HA (D175A), -hB3GALT4-D176A-3HA (D176A), -hB3GALT4-D177A-3HA (D177A), and -hB3GALT4-D291A-3HA (D291A). Right: Quantitative data of MFI from at least four independent experiments (mean ± SD, $n \geq 4$). $P$ values are from one-way ANOVA followed by Dunnett's test for multiple comparisons to control (WT). See also Supplementary Fig. 4b–e. Source data for (**b**) and (**h**) are provided as a Source Data file.

CD59[32], we transfected His-FLAG-GST-FLAG-tagged CD59 (HFGF-CD59) into wild-type and B3GALT4-KO HEK293 cells (B3GALT4-KO confirmation shown in Supplementary Fig. 2b). HFGF-CD59 was liberated from the cell surface by phosphatidylinositol-specific phospholipase C (PI-PLC), affinity-purified and analyzed. According to previous structural analyses of GPIs from different GPI-APs, eight glycoforms were expected to be found for mammalian GPIs (Fig. 3e). Since the most C-terminal trypsin cleavage site was double lysine (KK), the N-terminus of the peptide portion was either DL or KDL. Although GPI lacking GalNAc and Man4 side-chains (glycoform 1) and sialylated GPI with Man4 (glycoform 8) were not found, six glycoforms were detected in the wild-type cells (Supplementary Table 2 and Supplementary Fig. 3a–g). About 70% had a non-galactosylated GalNAc side-chain (glycoform 2, Fig. 3f, blue area in left bar) and 26% had GalNAc and one hexose, which could be Gal or Man4 (glycoform 3 or 6, orange). Approximately 1% had a galactosylated GalNAc side-chain and Man4 (glycoform 7, green) and another approximately 1% had a sialylated GalNAc side-chain (glycoform 4, purple). As expected, these two GPIs bearing Gal disappeared after KO of *B3GALT4* (Fig. 3f, second left bar and Supplementary Table 2). As also expected, GPI bearing non-galactosylated GalNAc side-chain increased from 70% to 81% after KO of *B3GALT4* (glycoform 2 or 6, blue), however, 18% of the GPI still had one hexose and a GalNAc side-chain (glycoform 3 or 6, orange). Because MS analysis does not differentiate Gal from Man, we next used B3GALT4-PIGZ-DKO HEK293 cells to eliminate Man4. Nearly 99% of GPIs from B3GALT4-PIGZ-DKO cells had only a non-galactosylated GalNAc side-chain (glycoform 2, middle bar). When B3GALT4-PIGZ-DKO cells were transfected with *B3GALT4* cDNA (Supplementary Fig. 2c, d for confirmation of transfection), glycoform 2 disappeared almost completely and most GPI had at least one hexose (Gal) (right bar). Therefore, B3GALT4 was essential for Gal-modification of GPI-GalNAc of GPI-APs. GPI from B3GALT4-transfected B3GALT4-PIGZ-DKO cells consisted of approximately 67% of the GPI-GalNAc bearing Gal or Man4, 9% bearing Gal and Man4 (glycoform 7), and 20% bearing Gal-Sia without (glycoform 4) or with Man4 (glycoform 8, Supplementary Fig. 3h) (Fig. 3f, right bar), indicating that overexpression of B3GALT4 by transfection caused overexpression of Gal-modification activity. Transfection of empty vector into B3GALT4-PIGZ-DKO cells resulted in occurrence of about 18% of hexose (Man4)-bearing GPI (orange in second right bar). These results indicated that the Man4-negative phenotype caused by PIGZ-KO was partially reverted after culture and transfection (see "Discussion" for a possible reason).

**B3GALT4 catalytic sites for galactosylation of GM2 and GPI.** Although B3GALT4 was identified as GM1 synthase more than 20

years ago, amino acid residues critical for enzyme activity have not been analyzed. To determine whether the same catalytic region is used for GM1 synthesis and GPI-GalNAc galactosylation, we aimed to identify critical residues for the enzymatic activities. Phylogenetic analysis suggested that various B3GALT proteins belong to at least three groups: B3GALT4, B3GALT6 and three other B3GalTs (B3GALT1–2 and B3GALT5) (Supplementary Fig. 4a). A homology model generated for human B3GALT4 using I-TASSER[33] suggested that a GT-A fold structure composed of a seven stranded β-sheet core flanked by α-helices and a small antiparallel β-sheet (Supplementary Fig. 4d), similar to typical Golgi GTs. Generally, GT-A enzymes contain a DXD motif which captures a divalent cation, usually $Mn^{2+}$, for binding the nucleotide-sugar. A DXD motif, such as DDD (residues 175–177) in human B3GALT4, is conserved among B3GALT4s (Supplementary Fig. 4b). To determine whether this DDD sequence is a functional DXD motif, we constructed B3GALT4 with mutations D175A, D176A or D177As. In addition, the conserved D291 of B3GALT4 is structurally equivalent to the catalytic residues E281 in glucuronyltransferase I[34] and D232 in mouse Manic Fringe (Supplementary Fig. 4c)[35]. Since D291 was predicted to locate closely to the DXD motif (Supplementary Fig. 4e), D291A mutant was also constructed. To perform functional assignment study for B3GALT4, we transfected B3GALT4 wild-type and mutant constructs into PIGS-B3GALT4-DKO cells (Fig. 3g). Overexpression of B3GALT4-D175A, -D176A, -D177A, and -D291A did not rescue galactosylation deficiency of GM2 or GPI-GalNAc (Fig. 3h), suggesting that DDD (residues 175–177) and D291 are required for B3GALT4 activities toward both GSL and GPI substrates, and D291 might function as the catalytic base of B3GALT4. These results show that the same catalytically important amino acids are used for both GM1 synthesis and GPI side-chain galactosylation.

**LacCer biosynthesis is required for galactosylation of GPI.** Knockout of *UGCG* induced positive T5 mAb staining of PIGS-KO HEK293 cells without affecting GS-II lectin staining (Fig. 2b). UGCG transfers glucose to ceramide to synthesize glucosylceramide (GlcCer) (schematic in Fig. 4a). We asked whether enzymatically active UGCG is required for GPI galactosylation. UGCG active site mutant D144A completely lost the ability to generate GlcCer[36]. PIGS-UGCG-DKO cells were stably transfected with wild-type UGCG or D144A mutant UGCG (Fig. 4b) to see the effects of the mutation on the induction of positive T5 mAb staining. Both GM1 synthesis and GPI-Gal modification were rescued by wild-type UGCG but not by the D144A mutant (Fig. 4c), indicating that enzymatic activity of UGCG is required for galactosylation of GPI-GalNAc. To determine whether the requirement for UGCG for galactosylation of GPI-GalNAc is a general phenomenon, we used CHO cells because we previously reported that CHO cells express free GPI bearing a galactosylated

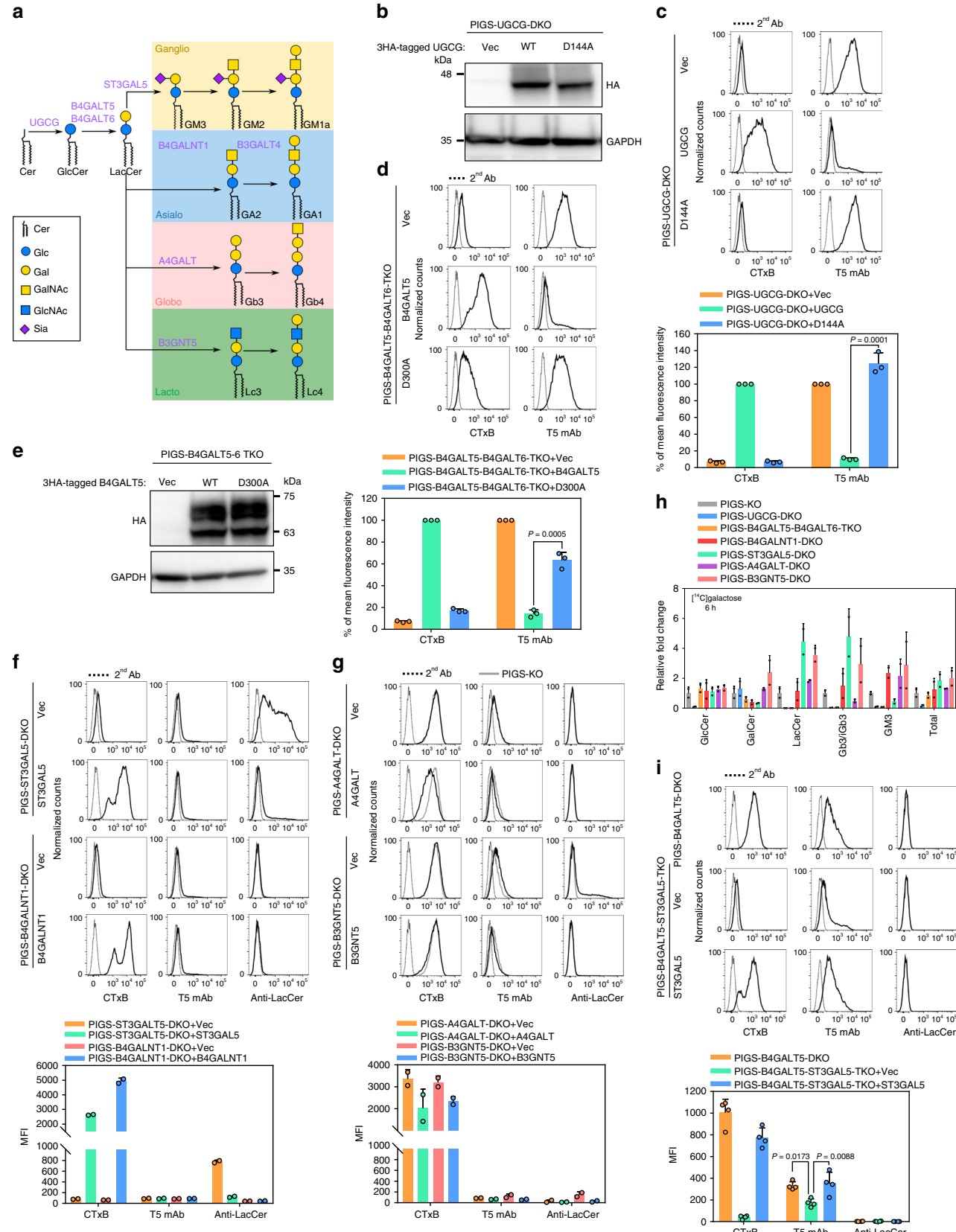

GalNAc side-chain as well as GPI-APs[5]. Knockout of *UGCG* from CHO cells also made T5-staining positive without affecting the GS-II lectin staining profile (Supplementary Fig. 5a). Therefore, the requirement for UGCG for GPI galactosylation seems to be a general phenomenon.

GlcCer is usually used for LacCer synthesis by LacCer synthase (Fig. 4a). Perhaps owing to the redundancy of LacCer synthases, our CRISPR screen did not suggest LacCer involvement in GPI galactosylation. B4GALT5 (β1,4-galactosyltransferase 5) and B4GALT6 (β1,4-galactosyltransferase 6) are the major and minor

**Fig. 4 Biosynthesis of LacCer is required for Gal modification of GPI-GalNAc. a** Schematic of GSL pathway. Biosynthesis of Ganglio and Asiaio are shown. **b** Western blotting of 3HA tagged UGCG. Lysates of PIGS-UGCG-DKO cells stably expressing pLIB2-BSD-3HA (Vec), -UGCG-3HA (UGCG), and UGCG-D144A-3HA (D144A) were analyzed. GAPDH, a loading control. **c** Top: PIGS-UGCG-DKO cells stably expressing Vec, UGCG or D144A were stained with CTxB and T5 mAb. Bottom: Quantitative data of MFI from three independent experiments (mean ± SD, $n = 3$). *P* values are from *t* test (unpaired and two-tailed) with comparisons to UGCG control. **d** Top: PIGS-B4GALT5-B4GALT6-TKO cells stably expressing Vec, B4GALT5 or D300A were stained with CTxB and T5 mAb. Bottom: Quantitative data of MFI from three independent experiments (mean ± SD, $n = 3$). *P* values are from *t* test (unpaired and two-tailed) with comparisons to B4GALT5 control. **e** Western blotting of 3HA tagged B4GALT5. Lysates of PIGS-B4GALT5-B4GALT6-TKO cells stably expressing pLIB2-BSD-3HA (Vec), -B4GALT5-3HA (B4GALT5), and -B4GALT5-D300A-3HA (D300A) were analyzed. **f** Top: PIGS-ST3GAL5-DKO cells transiently expressing Vec or mouse ST3GAL5 and PIGS-B4GALNT1-DKO cells transiently expressing Vec or human B4GALNT1 were stained with CTxB, T5 mAb, and anti-LacCer mAb. Bottom: Quantitative data of MFI from two independent experiments (mean ± SD, $n = 2$). **g** Top: PIGS-A4GALT-DKO cells stably expressing Vec or human A4GALT and PIGS-B3GNT5-DKO cells stably expressing Vec or human B3GNT5 were stained with CTxB, T5 mAb, and anti-LacCer mAb. Bottom: Quantitative data of MFI from two independent experiments (mean ± SD, $n = 2$). See also Supplementary Fig. 5c and d. **h** Quantitative data of high-performance thin-layer chromatography profile of [$^{14}$C] galactose-labeled GSLs from HEK293 cells. Data from two independent experiments (mean ± SD, $n = 2$). See also Supplementary Fig. 5e. **i** Top: PIGS-B4GALT5-ST3GAL5-TKO cells transiently expressing Vec or mouse ST3GAL5 and PIGS-B4GALT5-DKO cells were stained with CTxB, T5 mAb, and anti-LacCer mAb. Bottom: Quantitative data of MFI from four independent experiments (mean ± SD, $n = 4$). *P* values are from one-way ANOVA followed by Dunnett's test for multiple comparisons to Vec. Source data for (**c**, **d**), (**f**–**h**), and (**i**) are provided as a Source Data file.

LacCer synthases, respectively[37,38]. We generated PIGS-B4GALT5-DKO and PIGS-, B4GALT5-, and B4GALT6-triple knockout (TKO) cells. Knockout of *B4GALT5* reduced CTxB staining to less than 30% of the level in PIGS-KO cells and, at the same time, T5 mAb staining was increased (Supplementary Fig. 5b). Further knockout of *B4GALT6* abolished CTxB staining and T5 mAb staining further increased (Fig. 4d, top panel). An EDDD motif, which is conserved among B4GALTs, is thought to be essential for activity of enzymes in the B4GALT family[39]. Thus, a B4GALT5 D300A mutant was constructed and transiently or stably overexpressed in PIGS-B4GALT5-DKO cells and PIGS-B4GALT5-B4GALT6-TKO cells (Fig. 4e). The D300A mutant of B4GALT5 failed to rescue GPI galactosylation (Fig. 4e and Supplementary Fig. 5b), suggesting that LacCer synthase activity is also required for GPI galactosytlaion.

LacCer is present in the cells as an end product and is also converted to various GSLs (Fig. 4a). ST3GAL5 transfers Sia to LacCer to generate GM3; B4GALNT1 transfers GalNAc to LacCer and GM3 to generate GA2 and GM2, respectively; A4GALT transfers Gal to LacCer to generate Gb3; and B3GNT5 transfers GlcNAc to generate Lc3 (Fig. 4a)[40,41]. We knocked out these genes from PIGS-KO cells to determine whether GSLs more complex than LacCer are required for GPI galactosylation. Knockout of *ST3GAL5* from PIGS-KO cells increased the amount of LacCer and abolished GM1, while T5 mAb staining was not increased (Fig. 4f, top). Knockout of *B4GALNT1* from PIGS-KO cells abolished GM1 and failed to increase T5 mAb staining (Fig. 4f, bottom). These results suggest that GPI galactosylation does not require GM3 or GA2. Knockout of *A4GALT* or *B3GNT5* from PIGS-KO cells did not increase T5 mAb staining (Fig. 4g and Supplementary Fig. 5c, d), indicating that neither Gb3 nor Lc3 is required for GPI galactosylation. Analysis of GSL profiles of PIGS-KO cells defective in UGCG, B4GALT5/6, B4GALNT1, ST3GAL5, A4GALT or B3GNT5 showed that LacCer was lost by UGCG-KO and B4GALT5/6-KO whereas LacCer levels were similar after B4GALNT1-KO or increased after ST3GAL5-KO, A4GALT-KO or B3GNT5-KO (Fig. 4h and Supplementary Fig. 5e). Therefore, GPI galactosylation deficiency was correlated with a lack of LacCer. The partially reduced GPI galactosylation in PIGS-B4GALT5-DKO cells caused by knockout of the major LacCer synthase, as shown by appearance of T5 staining (Fig. 4i, upper), was partially rescued by further knockout of *ST3GAL5*, which might be due to the increase of LacCer in the Golgi, although LacCer present as an end product was not observed on the cell surfaces of PIGS-B4GALT5-ST3GAL5-TKO (Fig. 4i).

These results suggest that biosynthesis of LacCer is required for Gal modification of GPI-GalNAc.

Accumulation of Cer and sphingomyelin were previously observed in both UGCG-KO mice and B4GALT5-B4GALT6-DKO mice[42,43], but not in ST3GALT5-KO mice and B4GALNT1-KO mice[44,45]. We tested whether accumulation of Cer is the reason for the GPI galactosylation deficiency in cells lacking LacCer by using myoriocin and fumonisin B1, inhibitors of the Cer synthesis pathway[46,47] (Supplementary Fig. 5f). After treatments with these inhibitors, CTxB staining of PIGS-KO cells was reduced to 50% of levels of solvent-treated cells (Supplementary Fig. 5g), suggesting they inhibited Cer generation. T5 mAb staining of PIGS-UGCG-DKO and PIGS-B4GALT5-B4GALT6-TKO cells was not decreased after treatment with myriocin or fumonisin B1 (Supplementary Fig. 5h), suggesting that B3GALT4 activity was not increased upon myriocin or fumonisin B1 treatment. These results ruled out the possibility that accumulation of precursors like Cer caused the deficiency of GPI galactosylation in these cells. Taken together, our data show that the presence of LacCer is required for B3GALT4 to act on GPI-GalNAc.

**Lack of LacCer does not affect expression of B3GALT4.** To investigate how the loss of LacCer impairs GPI galactosylation, we next asked whether expression and/or localization of B3GALT4 is impaired in cells lacking LacCer. We first confirmed that loss of LacCer did not cause a decrease in *B3GALT4* mRNA levels (Supplementary Fig. 6a). We then asked whether endogenous B3GALT4 was maintained in LacCer-deficient cells. For this, we measured GM1 synthase activity in PIGS-KO, PIGS-B3GALT4-DKO, PIGS-UGCG-DKO, and PIGS-B4GALT5-B4GALT6-TKO cells in vitro, and confirmed that endogenous activity of B3GALT4 was not affected by LacCer loss (Supplementary Fig. 6b). We then determined the subcellular localization of B3GALT4. Because of the extremely low expression of B3GALT4 in cultured cell lines (Supplementary Fig. 6c), we could not detect endogenous B3GALT4 using commercial antibodies. Instead, we expressed FLAG-6His-tagged human B3GALT4, 3HA-tagged human PGAP4 (GPI-GalNAc transferase) and B4GALNT1 (GM2/GD2 synthase) in HeLa cells, and studied them by conventional confocal fluorescence microscopy-based linescan analysis. Intra-Golgi localizations of these proteins were determined by measuring the fluorescence profile along the linescan relative to the cis-Golgi and TGN markers GM130 and TGN46, respectively[48]. Consistent with previous reports

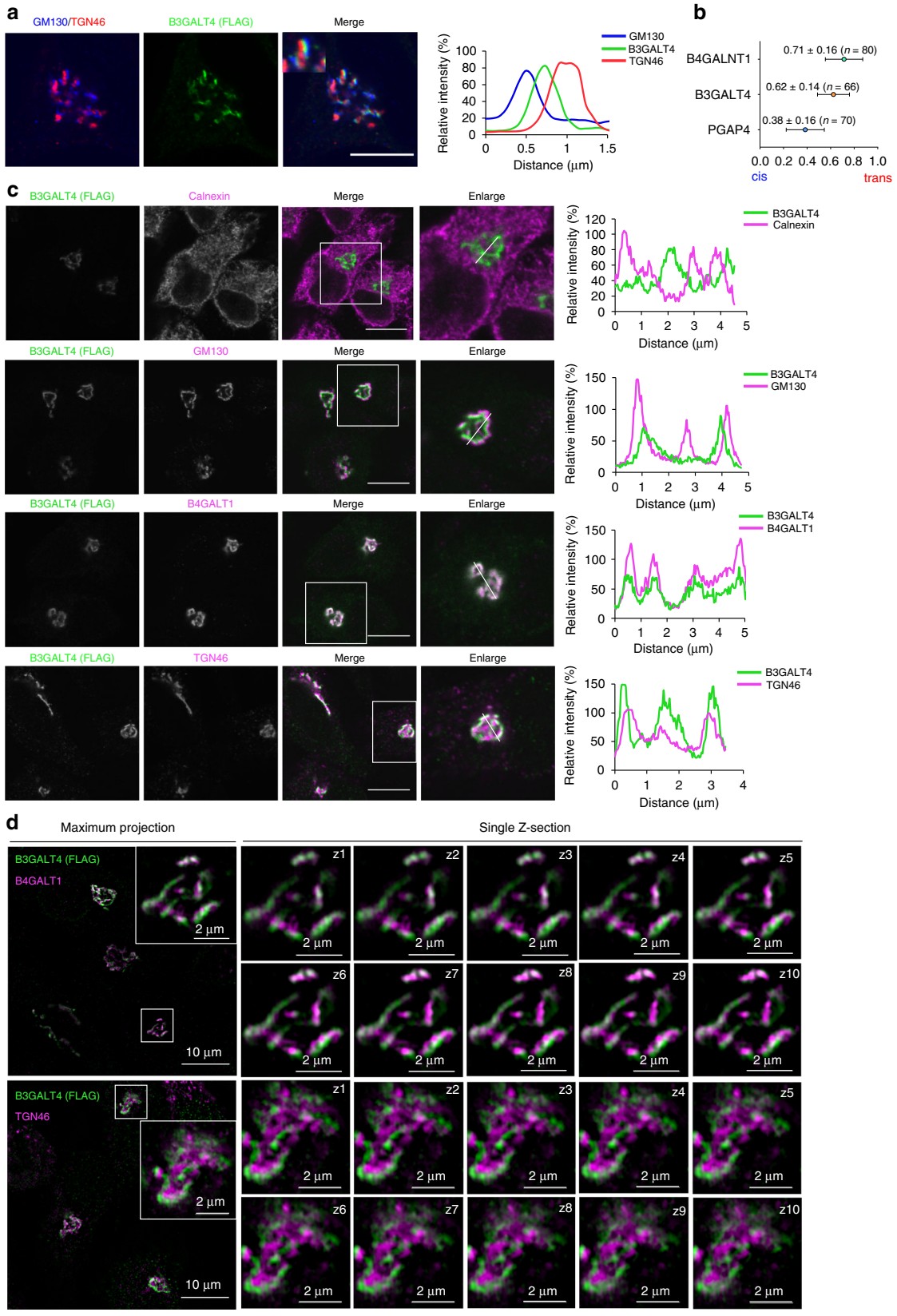

suggesting a Golgi-cisterna localization of mouse B3galt4[49] and a TGN localization of B4GALNT1[50], we observed that human B3GALT4 and B4GALNT1 localized to a late Golgi compartment, while PGAP4 localized to the cis- or medial-Golgi compartment (Fig. 5a, b and Supplementary Fig. 6d). We also found that

B3GALT4 was co-localized with B4GALT1 and partially co-localized with TGN46 (Fig. 5c). We then used super-resolution 3D structured illumination microscopy to compare localization patterns of FLAG-tagged B3GALT4 and endogenous markers for trans-Golgi cisterna (B4GALT1) and TGN (TGN46), and

**Fig. 5 B3GALT4 localizes at trans-cisternae of the Golgi. a** Left: HeLa cells stably expressing FLAG-6His tagged B3GALT4 were fixed and stained as described to detecting the FLAG tag (green) and a cis-Golgi marker (GM130; blue) and a TGN marker (TGN46; red). Scale bar, 10 μm. **b** Average localizations of the indicated Golgi proteins in HeLa cells (see Supplementary Fig. 6d) relative to GM130 (cis) and TGN46 (TGN) markers. Error bars indicate mean ± SD the number of measurements by linescan analysis for PGAP4 (41 images), B4GALNT1 (31 images), and B3GALT4 (29 images). **c** B3GALT4 colocalizes with the Golgi markers in HeLa cells. Left: Cells stably expressing FLAG-6His tagged B3GALT4 were fixed and stained for FLAG tag (green), as well as markers (magenta): Calnexin (ER), GM130 (cis-Golgi), B4GALT1 (Trans Golgi cisterna), and TGN46 (TGN). Scale bar, 10 μm. **d** Localization of B3GALT4 analyzed with 3D-SIM super-resolution microscopy. Left: a maximum intensity z-projection of B3GALT4 (green) and B4GALT1 or TGN46 (magenta), and a magnified region of Golgi in one of the cells. Right: ten magnified serial z-sections. For (**a**) and (**b**), right panel shows linescan analysis along a portion of the white line overlaying the image (left). Source data for (**b**) are provided as a Source Data file.

observed spatial separation of B3GALT4 and TGN46 but overlap of B3GALT4 and B4GALT1 (Fig. 5d). Taken together, our results indicate a trans-Golgi cisterna localization of B3GALT4 in wild-type HeLa cells.

To understand whether loss of UGCG affects correct localization of B3GALT4, we examined the expression level and subcellular localization of FLAG-6His-tagged B3GALT4 stably expressed in wild-type and UGCG-KO HeLa cells (Supplementary Fig. 6e for KO confirmation). B3GALT4 expression and the N-glycan profile (sensitivity to Endo H digestion[49]) were not altered upon UGCG knockout (Supplementary Fig. 6f). B3GALT4 remained in the Golgi cisterna in UGCG-KO HeLa cells, but its localization was shifted to the ER by brefeldin A (BFA) treatment (Supplementary Fig. 6g). These results indicate that expression and intra-Golgi localization of B3GALT4 was not affected by the absence of LacCer.

**LacCer enhances B3GALT4 activity toward GPI-GalNAc.** We next asked whether LacCer is a prerequisite for B3GALT4 activity toward GPI-GalNAc. 3HA-tagged B3GALT4 was transiently expressed in PIGS-B3GALT4-DKO, PIGS-UGCG-DKO, and PIGS-B4GALT5-B4GALT6-TKO cells, and B3GALT4 expression was confirmed by Western blotting (Fig. 6a). In addition, B3GALT4 could be detected in the Golgi of cells defective in UGCG or B4GALT5 and B4GALT6 (Fig. 6b). T5 mAb staining levels of PIGS-UGCG-DKO and PIGS-B4GALT5-B4GALT6-TKO cells were decreased by transient overexpression of 3HA-tagged B3GALT4 by about 40%, suggesting that highly expressed B3GALT4 has activity toward GPI-GalNAc even without LacCer in vivo (Fig. 6c). The reduction of T5 mAb staining levels in the absence of LacCer was milder than the reduction in the presence of LacCer (PIGS-B3GALT4-DKO cells): with comparable expression levels of B3GALT4, the remained T5 mAb staining was 2.50 ± 0.38- and 2.75 ± 0.06-times that of PIGS-UGCG-DKO and PIGS-B4GALT5-B4GALT6-TKO cells compared with PIGS-B3GALT4-DKO cells transfected with B3GALT4 cDNA (Fig. 6c, right). To see the rescue of GPI galactosylation by reduction of T5 mAb staining levels, transient expression system seemed to be inefficient because even when newly synthesized GPIs are galactosylated, pre-existing non-galactosylated GPIs are positively stained by T5 mAb. To avoid such situation, we next used a stable expression system. Stable overexpression of B3GALT4 rescued galactosylation deficiency caused by loss of GlcCer or LacCer generation very clearly (Fig. 6d, e). These results suggest that overexpression of B3GALT4 in LacCer-null cells can rescue galactosylation deficiency less efficiently. Taken together, LacCer is not required for GPI galactosylation when B3GALT4 is abundant; however, LacCer might greatly enhance the efficiency of GPI galactosylation by endogenous B3GALT4 (Fig. 6f), which is usually present at a low level in most cells (Supplementary Fig. 6c).

**Biosynthesis of GPI is regulated by ERAD in PIGS-KO cells.** ERAD, a multicomponent system initially found as a mechanism

for degradation of unfolded/misfolded secretory proteins, is also involved in negative regulation of a number of ER-resident proteins[51,52]. Knockout of three ERAD genes, SYVN1, UBE2J1, and UBE2G2, from PIGS-KO cells led to positive staining by T5 mAb (Fig. 2b). HRD1, encoded by SYVN1, is a major E3 ubiquitin ligase in the ERAD system, and UBE2J1 and UBE2G2 are E2 enzymes known to work with HRD1[28,53]. To further characterize ERAD involvement in the regulation of GPI biosynthesis, we first determined the contribution of other ERAD components. Knockout of SEL1L and DERL2, two components known to work with HRD1[54,55], resulted in positive T5 mAb staining of PIGS-KO cells (Fig. 7a). Our genome-wide screen data also showed that one of the sgRNAs for DERL2 was highly enriched (Supplementary Data 2 and 3). Knockout of DERL1, a DERL2 homologue, had only a mild effect. Knockout of OS9 and even double knockout of OS9 and XTP3B, encoding lectins involved in N-glycan dependent recognition of ERAD substrates[56], did not result in positive T5 mAb staining, indicating that recognition is N-glycan-independent (Fig. 7a). Knockout of another ERAD E3 ligase, AMFR/gp78, caused clear induction of positive T5 mAb staining (Fig. 7a). These results indicate that regulation of GPI biosynthesis by ERAD is mainly mediated by the HRD1-dependent pathway; that the gp78-dependent pathway may also be involved; and that recognition of the substrate(s) is N-glycan-independent.

We then sought the mechanism of how ERAD disruption caused positive T5 mAb staining in PIGS-KO cells. When ERAD genes were disrupted, GM1 levels were almost unchanged (Fig. 2b), suggesting that B3GALT4 activity was not influenced by ERAD disruption. Because positive T5 mAb staining could also be caused by increased synthesis of free GPI-GalNAc surpassing the galactosylation capacity of B3GALT4, we determined the levels of free GPI-GalNAc in PIGS-KO cells in the presence and absence of ERAD. To compare the total amounts of GPI-GalNAc, we tested in SLC35A2-KO conditions where UDP-Gal, a Gal donor, is not available in the Golgi (Supplementary Fig. 7 for confirmation of SLC35A2 KO). The T5 mAb staining intensity of PIGS-UBE2J1-SLC35A2-TKO cells was approximately seven times that of PIGS-UBE2J1-DKO cells (Fig. 7b). An increase of free GPI-GalNAc by UBE2J1-KO was also shown by western blotting (Fig. 7c). These results suggest that high production of free GPI-GalNAc exceeded the galactosylation capacity of endogenous B3GALT4. To confirm this mechanism, we transfected B3GALT4 cDNA into ERAD-defective PIGS-KO cells, such as PIGS-UBE2J1-DKO cells. The overexpression of B3GALT4 by transfection in fact decreased T5 mAb staining intensities of ERAD-defective PIGS-KO cells (Fig. 7d).

To test whether the increase of free GPI-GalNAc in ERAD-defective PIGS-KO cells was due to increased GPI biosynthesis, we metabolically labeled cells with [2-³H] mannose and analyzed radiolabeled GPI mannolipids by thin-layer chromatography using a phosphorimager. PIGS-KO CHO cells were used to identify the spots of GPI biosynthetic intermediates (H5 and H6) and mature forms (H7, H7' and H8) based on their mobility[57].

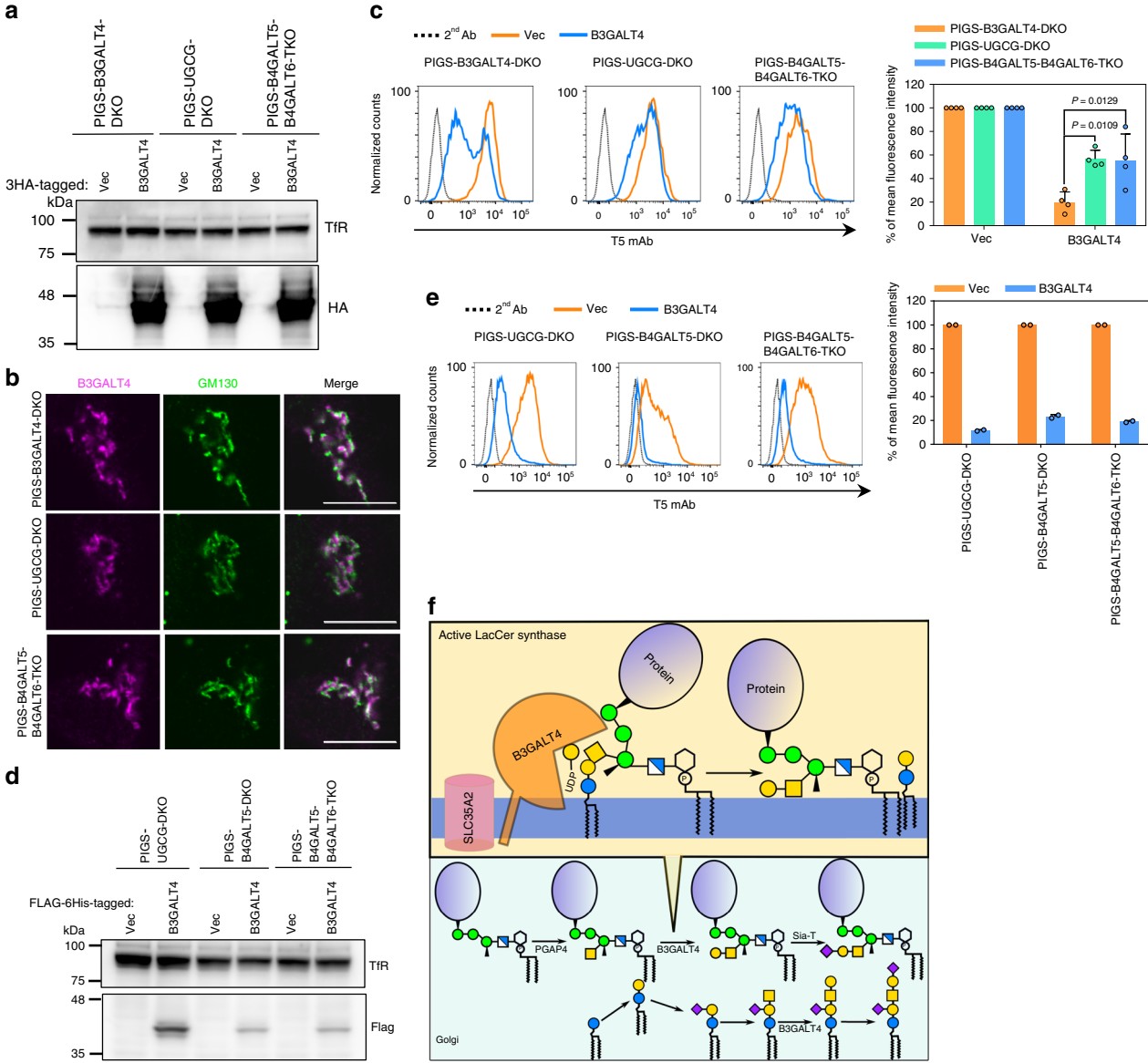

**Fig. 6 LacCer enhances B3GALT4 activity toward GPI-GalNAc. a** Western blotting of B3GALT4-3HA transiently overexpressed in PIGS-UGCG-DKO, PIGS-B4GALT5-DKO, and PIGS-B4GALT5-B4GALT6-TKO cells. **b** Representative fluorescence images of 3HA-tagged B3GALT4 in PIGS-B3GALT4-DKO, PIGS-UGCG-DKO, and PIGS-B4GALT5-B4GALT6-TKO HEK293 cells. GM130, a marker for cis-Golgi. Scale bar, 10 μm. **c** Left: PIGS-UGCG-DKO, PIGS-B4GALT5-DKO, and PIGS-B4GALT5-B4GALT6-TKO cells transiently expressing pME-B3GALT4-3HA were stained with T5 mAb. Right: Quantitative data of MFI from four independent experiments (mean ± SD, n = 4). P values are from one-way ANOVA followed by Dunnett's test for multiple comparisons to PIGS-B3GALT4-DKO cells. **d** Western blotting of FLAG-6His tagged B3GALT4. Lysates of PIGS-UGCG-DKO, PIGS-B4GALT5-DKO, and PIGS-B4GALT5-B4GALT6-TKO cells stably expressing empty vector (Vec) and B3GALT4 were analyzed. TfR, a loading control. **e** Left: PIGS-UGCG-DKO, PIGS-B4GALT5-DKO, and PIGS-B4GALT5-B4GALT6-TKO cells stably expressing B3GALT4 were stained with T5 mAb. Right: Quantitative data of MFI from two independent experiments (mean ± SD, n = 2). **f** Schematic of LacCer enhanced galactose modification on GPI-GalNAc by B3GALT4 in the Golgi. Source data for (**c**) and (**e**) are provided as a Source Data file.

GPI biosynthesis was greatly increased in ERAD-defective PIGS-KO HEK293 cells compared to ERAD-active PIGS-KO HEK293 cells (Fig. 7e), suggesting that GPI biosynthesis is strongly suppressed by ERAD in PIGS-KO cells. In contrast, the enhancement of GPI biosynthesis was not observed when SYVN1 (HRD1) or UBE2J1 was knocked out in SLC35A2-KO HEK293 cells, in which GPI-Tase is intact (Fig. 7f, g). These results suggest that when GPI attachment to proteins is defective, ERAD strongly suppressed GPI biosynthesis to prevent accumulation of non-protein-linked GPIs and its biosynthetic intermediates.

To understand mechanistic basis of GPI biosynthesis upregulation in ERAD-defective PIGS-KO cells, we used a microarray

analysis to compare mRNA levels of GPI biosynthesis genes between PIGS-KO and PIGS-UBE2J1-DKO cells. Although transcript levels of many genes were affected by *UBE2J1* knockout (Supplementary Fig. 7b), no significant changes in transcript levels of GPI biosynthesis genes were observed (Fig. 7h). Thus, upregulation of GPI biosynthesis in PIGS-UBE2J1-DKO cells is not due to upregulation of GPI biosynthesis pathway genes.

Because it is likely that precursors of GPI-APs are degraded by the ERAD system in PIGS-KO cells, we next tested a possibility that precursors of some GPI-APs might accumulate and trigger the enhancement of GPI biosynthesis in ERAD-defective PIGS-KO cells. For this, we transiently overexpressed two GPI-APs,

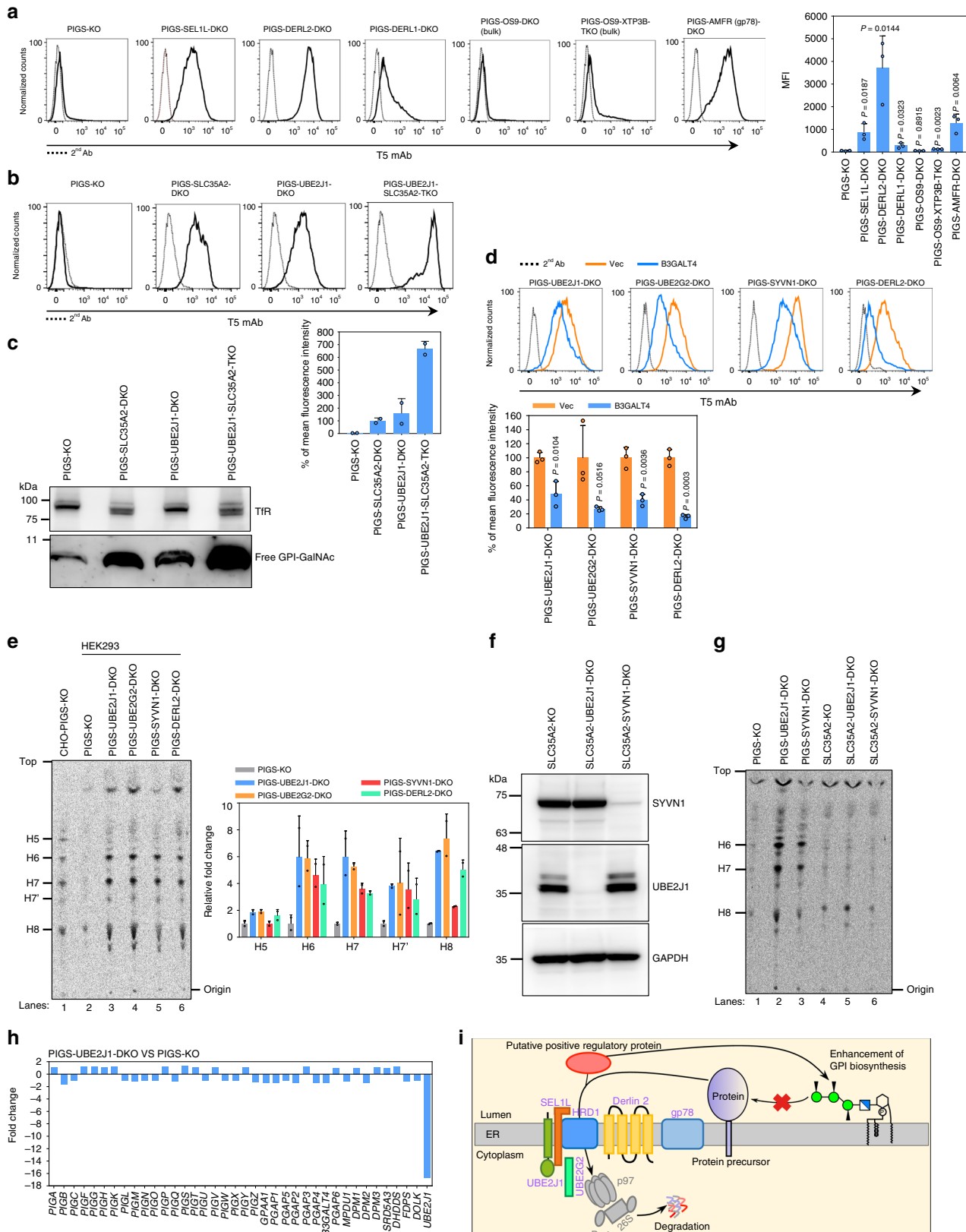

CD59 and Prion (PrP), in PIGS-KO and PIGS-UBE2J1-DKO cells (Supplementary Fig. 8a), and asked whether accumulation of these GPI-APs' precursors occurs and triggers upregulation of GPI biosynthesis. After transfection of FLAG-tagged CD59 cDNA under overexpression conditions, CD59 precursor was undetectable in PIGS-KO cells but was strongly detected in PIGS-

UBE2J1-DKO cells (Supplementary Fig. 8b). CD59-lacking the C-terminal signal (CD59-del-C) was strongly detected in both cell types. Therefore, CD59 precursor bearing the C-terminal signal was efficiently degraded by ERAD, and accumulation was not achieved by transfection. In case of PrP, the precursor bearing the C-terminal signal was detectable in PIGS-KO cells and was

**Fig. 7 Biosynthesis of GPI is under regulation by ERAD in PIGS-KO cells. a** Left: Flow cytometry analysis of ERAD gene knockout in PIGS-KO cells. Right: Quantitative data of MFI from three independent analyses (mean ± SD, $n = 3$). P values are from $t$ test (unpaired and two-tailed) with comparisons to control (PIGS-KO). **b** Left: Flow cytometry analysis of PIGS-KO, PIGS-SLC35A2-DKO, PIGS-UBE2J1-DKO and PIGS-UBE2J1-SLC35A2-TKO HEK293 cells stained by T5 mAb. Right: Quantitative data of MFI from two independent experiments (mean ± SD, $n = 2$). See also Supplementary Fig. 7. **c** Western blotting of free GPI-GalNAc in ERAD-deficient cells. Lysates of PIGS-KO, PIGS-SLC35A2-DKO, PIGS-UBE2J1-DKO and PIGS-UBE2J1-SLC35A2-TKO cells were analyzed with T5 mAb. TfR, a loading control. **d** Top: PIGS-UBE2J1-DKO, PIGS-UBE2G2-DKO, PIGS-SYVN1-DKO and PIGS-DERL2-DKO HEK293 cells transiently expressing Vec or B3GALT4 were stained by T5 mAb. Bottom: Quantitative data of MFI from three independent experiments (mean ± SD, $n = 3$). P values are from $t$ test (unpaired and two-tailed) with comparisons to Vec control. **e** GPI biosynthesis. Left: Cells were metabolically labeled with [$^3$H] mannose, GPI intermediates were extracted and analyzed by HPTLC. H5, GPI with one Man and one EtNP; H6, GPI with three Mans; H7, H7′, and H8, mature forms of GPI. Right: Quantitative data from two independent experiments (mean ± SD, $n = 2$). **f** Confirmation of KO of UBE2J1 or SYVN1 in SLC35A2-KO HEK293 cells by Western blot. GAPDH, a loading control. **g** GPI biosynthesis of ERAD-deficient SLC35A2-KO HEK293 cells. **h** Microarray of PIGS-KO and PIGS-UBE2J1-DKO HEK293 cells. Expression level changes of 36 genes of GPI pathway are shown. The UBE2J1 gene was significantly downregulated in the PIGS-UBE2J1-DKO cells, probably caused by nonsense-mediated mRNA decay. See also Supplementary Fig. 7b and Supplementary Data 4. **i** ERAD-deficiency enhances GPI biosynthesis. Genes validated are shown in light purple. Source data for (**a**), (**b**), (**d**), and (**e**) are provided as a Source Data file.

detected more strongly in PIGS-UBE2J1-DKO cells, showing some accumulation was caused in PIGS-KO cells (Supplementary Fig. 8b). Even under these conditions with accumulated PrP precursor, T5 mAb staining of PIGS-KO cells was not changed (Supplementary Fig. 8c). To eliminate a possibility that the insensitive response to GPI biosynthesis level measured by T5 mAb staining was caused by galactosylation of GPI side-chain, we overexpressed PrP and PrP-del-C in PIGS-B3GALT4-DKO cells but again we did not observe any increase of T5 mAb staining level (Supplementary Fig. 8d). Therefore, accumulation of PrP precursors was not a trigger of GPI biosynthesis upregulation. The mechanisms of upregulation of GPI biosynthesis and its ERAD-mediated suppression in PIGS-KO cells should be clarified to better understand the quantitative regulation of GPI biosynthesis (Fig. 7i).

## Discussion

A major finding of this study is that a galactosyltransferase that attaches Gal to a side-chain of GPI is the same galactosyl-transferase—B3GALT4—involved in biosynthesis of glyco-sphingolipids GA1, GM1a, GD1b, and GT1c, thus demonstrating an unexpected relationship between the GPI and GSL biosyn-thetic pathways. We thought until this finding that the two bio-synthetic pathways were totally independent. Moreover, this study revealed that LacCer is necessary for galactosylation of GPI side-chains by B3GALT4 in the Golgi apparatus, demonstrating a functional relationship between two major glycolipid groups during the biosynthesis. We previously thought that interactions of GPI-APs and GSLs occurred only after their synthesis was complete. For example, GPI-APs and GSLs dynamically interact, generating membrane microdomains in the plasma membrane. Our findings demonstrate that the two biosynthetic pathways have a close functional interaction in the Golgi.

In line with previous observations of GPI structures from prions[8], we found that sialylation of GPI side-chains in His-FLAG-GST-FLAG-tagged CD59 is quite limited because GPI side-chain galactosylation is limited (Fig. 3f), and noticed that overexpression of B3GALT4 could increase GPI sialylation. Because B3GALT4 is used in both GSL and GPI biosynthesis pathways, they might share the next step enzyme, Sia-T. Also, we noticed that Man4 modification of GPI was once lost by PIGZ-KO from HEK293 cells (Fig. 3f, middle bar) but reappeared partially after culture (second right bar). This suggests that another mannosyltransferase generated Man4 in PIGZ-KO HEK293 cells because PIGZ-KO is presumably irreversible. PIGB is the candidate, because PIGB belongs to the same family of mannosyltransferases as PIGZ and transfers α1,2-Man to

Man3[58]. Moreover, PIGB from *Trypanosoma* and *Plasmodium* compensated Smp3 (PIGZ ortholog) deficiency in yeast[59].

We determined that GPI-galactosylation is regulated by LacCer production. Because LacCer is a common partial structure of the known acceptor substrates of B3GALT4, GA2, GM2 and GD2 (Fig. 4a), LacCer might directly bind to B3GALT4. Structural studies of B3GALT4 will clarify whether the conformation of B3GALT4 changes upon association with LacCer, which may explain the enhancement effect of LacCer for GPI galactosylation by B3GALT4.

BFA treatment of cells abolished GM2 and GM1 biosynth-esis[60,61], and a previous study suggested a TGN localization of B4GALNT1 (GM2 synthase)[50], and that physical interaction between B3GALT4 and B4GALNT1 might be the rate-limiting step in GM1 synthesis[62], supporting a TGN-localization of B3GALT4 for GM1 biosynthesis. However, our results and a previous study using BFA suggested that B3GALT4 was mainly localized at the Golgi cisterna[49]. GPI galactosylation requires LacCer, which is usually generated in the Golgi cisterna[61,63]. Our results suggest that B3GALT4 may function at the Golgi cisterna, which might be a reason for previous discrepant findings regarding the localization of B3GALT4 protein and GM1 bio-synthesis activity. Although GM3 is also generated in the Golgi cisterna[61,63], GM3 is not necessary for GPI galactosylation. Thus, B3GALT4 should modify GPI-GalNAc before generation of GM3, GM2, and GM1. However, it is still possible that GM1 is mainly generated in the Golgi cisterna. Since a study suggested that the inhibition of GM2 synthesis by BFA does not necessarily mean that B4GALNT1 is localized to the TGN, BFA could not be used to determine the location of GM2 or even GM1 synthesis[64].

Although neuroprotective properties of ganglioside GM1 have been demonstrated and accepted, no B3GALT4-deficient patients or mice have been reported[65]. One reason might be that B3GALT4-deficient mice have similar or even weaker phenotypes than those observed in B4GALNT1-deficient mice. Now that we have demonstrated that B3GALT4 also modifies GPI, B3GALT4-KO mice should be generated and studied in future. In addition, owing to the limitations of T5 mAb, current information about the distribution of free GPI-GalNAc within mouse tissues is incomplete[18]; generation of B3GALT4-KO mice is critical to determine whether free GPIs are more widely distributed.

Using free GPIs instead of GPI-APs as reporters in screening, we identified more regulators of GPI biosynthesis. We found that disruption of HRD1 and several other ERAD components enhanced GPI synthesis in GPI-Tase-deficient cells. How the ERAD pathway regulates GPI biosynthesis remains unclear. This could be specific to GPI-Tase-deficient cells, because GPI not used for generation of GPI-APs will accumulate. It is unlikely that

ERAD is involved in degradation of free GPIs because our data showed that GPI biosynthesis was upregulated when ERAD was defective. Cells might use ERAD for suppressing GPI synthesis through degradation of protein factors required for enhanced biosynthesis of GPI. Disruption of ERAD will impair such suppression, resulting in increased free GPIs in the GPI-Tase-deficient cells. More than 50 patients with inherited GPI deficiency caused by partial loss of GPI-transamidase have been reported[19]. In these patients' cells, GPI might be excessive to the transfer capacity of the GPI-transamidase, leading to accumulation of GPI if not properly regulated. Other possible situation to have excessive GPI might occur with ER stress where translation of secretory proteins including GPI-AP precursors is generally downregulated. Under ER stress, down regulation of GPI synthesis might also be required to prevent GPI accumulation. In contrast, normal cells express GPI-APs and limited free GPIs, therefore, ERAD might not be critically involved in regulating GPI biosynthesis to ensure sufficient, but not excessive, GPIs for production of GPI-APs. Our results indicated that increase of free GPIs in the absence of ERAD is caused by upregulation of biosynthesis (Fig. 7e). Because mRNA levels of genes involved in GPI biosynthesis were not changed (Fig. 7h), GPI biosynthesis must be upregulated by post-transcriptional mechanisms. Precursors of CD59 and PrP were accumulated in PIGS-UBE2J1-DKO cells (Supplementary Fig. 8b), suggesting that precursors of other GPI-APs are also degraded by ERAD. Whether precursors of some GPI-APs function as the trigger for enhancement of GPI biosynthesis pathway is still unclear. It will be of interest to understand how the ERAD system negatively regulates GPI biosynthesis.

## Methods

**Cells and culture**. HEK293 cells (ATCC CRL-1573) or HeLa cells (ATCC CCl-2) and their derivatives were cultured in Dulbecco's modified Eagle's medium (DMEM) with high glucose (Nacalai Tesque, Japan) containing 10% heat-inactivated Fetal Bovine Serum (FBS, Sigma). CHO K1 cells (ATCC CCL-61) and their derivatives were maintained in DMEM/F-12 (Nacalai Tesque) supplemented with 10% heat-inactivated FBS. All cells were maintained at 37 °C and in 5% $CO_2$.

**Antibodies and reagents**. Mouse monoclonal anti-*Toxoplasma gondii* GPI anchor (clone T5 4E10) antibody (T5 mAb) (1:100 for FACS or 1:1000 for Western blot or WB) was a generous gift from Dr. Jean François Dubremetz (Montpellier University, France)[17]. T5 mAb (# NR-50267) is now available from BEI Resources, NIAID, NIH. Biotin-conjugated Mouse mAb against human CD59 (clone 5H8) was previously described[66]. Mouse mAb against DYKDDDDK (FLAG) (# 014−22383) (1:1000 for WB; 1:500 for immunofluorescence or IF) was from Wako. Mouse mAbs against Transferrin Receptor (TfR) (# 13−6800) (1:1000 for WB), GAPDH (# MA1-22670) (1:1000 for WB), Lactosylceramide (CD17) (# MA1-10118) (1:100 for FACS), Alexa Fluor-conjugated goat against mouse IgG (1:500 for IF), rabbit IgG (1:500 for IF), Alexa Fluor 488-conjugated goat against mouse IgM (# A-10680) (1:500 for FACS or IF), HRP-conjugated goat against mouse IgM (# 62−6820) (1:1000 for WB), and Alexa Fluor-conjugated donkey against mouse IgG, rabbit IgG, and sheep IgG (1:500 for IF) were from Thermo Fisher Scientific. Rabbit mAb against GM130 (# ab76154) (1:250 for IF) and Alexa Fluor 647-conjugated goat against mouse IgM (1:500 for FACS) (# ab150123) were from Abcam. Mouse mAbs against UBE2J1 (# sc-377002) (1:500 for WB), UBE2G2 (# sc-393780) (1:500 for WB), GOSR1 (# sc-271551) (1:500 for WB), and Syntaxin 5 (# sc-365124) (1:500 for WB) were from Santa Cruz Biotechnology. HRP-conjugated goat against mouse IgG and rabbit IgG (1:1000 for WB) were from GE Healthcare. Rabbit pAb against SYVN1 (# 13473-1-AP) (1:1000 for WB) was from Proteintech. Rabbit pAb against B4GALT1 (# HPA010807) (1:500 for IF) was from Atlas Antibodies. Sheep pAb against TGN46 (# AHP500G) (1:500 for IF) was from Bio-Rad. Mouse mAb against GM130 (# 610822) (1:250 for IF) was from BD Biosciences. Rabbit mAb against HA (# 3724) (1:1000 for WB; 1:500 for IF) was from Cell Signaling Technology. Mouse mAbs against HA (# H3663) (1:1000 for WB), α-Tubulin (# T9026) (1:1000 for WB), and FLAG (FITC-conjugated) (# F4049) (1:100 for FACS) were from Sigma-Aldrich. Phycoerythrin (PE)-conjugated goat against mouse IgG (# 405307) (1:100 for FACS) was from Biolegend. Alexa Fluor 647-conjugated Griffonia Simplicifolia II (GS-II) (# L32451) (1:200 for FACS), Alexa Fluor 488-conjugated Cholera enterotoxin subunit B (CTxB) (# C34775) (1:200 for FACS), and Alexa Fluor 594-conjugated CTxB (# C22842) (1:200 for IF) were from Thermo Fisher Scientific. PE-conjugated Streptavidin (# 554061) (1:200 for FACS) was from BD Biosciences. PNGase F (# P0704) and Endo

Hf (# P0703) were from New England Biolabs. Brefeldin A (# 11861), myriocin (# 63150), and fumonisin B1 (# 62580) were from Cayman Chemical. Tunicamycin (# BIT1006) was from Wako. Uridine 5′-diphosphogalactose disodium salt (UDP-Gal) (# U4500) and monosialoganglioside GM2 (# G8397) were from Sigma-Aldrich.

**Characterization of PIGS-KO HEK293 cells for CRISPR screen**. A successful phenotype-driven, forward genetic screen relies on specific and stable phenotype. Before using PIGS-KO HEK293 cells in GeCKO screen for studying GPI galactosylation, we characterized the free GPI side-chain glycosylation using T5 mAb. The positive T5 mAb staining phenotype of PIGS-SLC35A2-DKO HEK293 cells was analyzed by PI-PLC treatment and further knockout of PIGO gene, essential for GPI biosynthesis. The T5 mAb staining was GPI-dependent, because the staining was reduced by PI-PLC or KO of PIGO (Supplementary Fig. 1b, c). In addition, the glycosylation of free GPI was further confirmed by a negative effect of overexpression of PGAP4 on T5 mAb staining (Supplementary Fig. 1d), suggesting both GalNAc modification and galactosylation of free GPIs in PIGS-KO HEK293 cells are efficient.

**Viral production and functional titration**. To produce lentivirus, the human GeCKOv2 pooled plasmids (lentiCRISPRv2) were co-transfected with lentiviral packaging pladmids pLP1, pLP2, and pLP/VSVG (Thermo Fisher) into Lenti-X 293T cells (Clontech). Lenti-X 293T cells were seeded at 70% confluence to one 10 cm-dish 8 h before transfection to reach about 90% confluence before transfection. Transfection was performed using PEI "Max" (Polyscience). For one 10 cm-dish, 6.3 μg pLP1, 2.1 μg pLP2, 3.2 μg pLP/VSVG and 8.4 μg lentiCRISPRv2 (4.2 μg library A and 4.2 μg library B) was diluted in 1.25 mL OptiMEM (Thermo Fisher). Forty microliters of PEI "Max" (2 mg/mL) was diluted in 1.25 mL OptiMEM, after 5 min incubation at room temperature (RT), this PEI "Max" reagent was added to the DNA reagent. 25 min later, the combined mixture was added to the cells. After 12 h, the media was changed to 10 mL pre-warmed DMEM supplemented with 10% FBS. About 24 h after change the media, the viral media was collected and 10 mL pre-warmed DMEM was added to the cells. The supernatant was filtered through a membrane (Mllex 0.45 μm, PVDF, 33 mm). Forty-eight and seventy-two hours after change the media, the viral media was collected and filtered. Finally, 30 mL viral media in total was combined and stored at 4 °C for functional titration and pooled screen as soon as possible.

For lentiviral functional titration, about $2.5 \times 10^5$ PIGS-KO HEK293 cells per well were plated in 6-well plates 36 h before adding the viral media. Different volumes of viral supernatant (25, 50, 100, 150, 200, 300, 400, and 500 μL) were added to each well (around $6 \times 10^5$ cells per well) along with a no virus control with 8 μg/mL polybrene (Sigma-Aldrich). Cells were incubated overnight (12–16 h), and media were changed to pre-warmed DMEM for all cells. At 24 h post-transduction, cells in all wells were split to a 1:6 ratio to prevent any well from confluence, and 10, 20, 25, 30, and 50% confluent controls were set at the same time. At 2 days and a half post-transduction, except for the confluent controls, new DMEM was supplemented with 0.5 μg/mL puromycin. Cells were split 1:6 to prevent any well from confluence. At 10 days post-transduction, all cells in the no virus control with puromycin were floating. The viral volume that results to 30% (MOI ≈ 0.3) cell survival in puromycin was determined by comparing with the 30% confluence controls.

**Pooled library transduction into PIGS-KO HEK293 cells**. For large-scale pooled screen, $3.5 \times 10^6$ PIGS-KO HEK293 cells were plated in one 15 cm-dish 36 h before adding the virus, $6 \times 15$ cm-dishes in total were prepared. Around $6 \times 10^7$ cells ($1 \times 10^7$ cells per dish) were transduced with 4.8 mL viral supernatant (more than 100 cells per lentiCRISPR construct). Cells were selected at 0.5 μg/mL puromycin, until the infected cells were expanded to $2.4 \times 10^8$ to keep the complexity of gRNA library, cells were combined and can be split 1:4, a minimum of $6 \times 10^7$ cells were placed for culture. At 2 weeks post-transduction, pellet of $5 \times 10^7$ cells without sorting was stocked at −80 °C. Approximately $1 \times 10^8$ cells were prepared for enrichment of T5 mAb staining-positive cells by FACSAria (BD).

**Enrichment of PIGS-KO cells positively stained by T5 mAb**. Around $1 \times 10^8$ cells were prepared for the first sorting. Briefly, cells were harvested and stained by T5 mAb, then followed by staining of anti-mouse IgM. After washing by PBS, cells were resuspended in PBS and sorted on a cell sorter. $2 \times 10^7$ cells were prepared for the second sorting and the third sorting. After sorting, the cells were maintained in DMEM supplemented with 0.25 μg/mL puromycin. Pellets of $2 \times 10^7$ sort3 cells were stored at −80 °C until use.

**sgRNA library readout by deep sequencing**. Approximately $5 \times 10^7$ control cells and $2 \times 10^7$ sort3 PIGS-KO HEK293 cells were used for genomic DNA extraction by Wizard Genomic DNA Purification Kit (Promega). Approximately 325 μg of genomic DNA from control cells and 30 μg of genomic DNA from sort3 cells were used for amplification of gRNA. PCR (25 cycles) was performed to amplify the sgRNAs using Ex Taq Polymerase (Takara), in total 65 tubes for control cells and 12 tubes for sort3 cells (Oligos for amplification of sgRNAs are in Supplementary Data 1). All the PCR products were combined and mixed well, and 675 μL PCR

products for control cells and 320 µL were applied to the 2% Gel for purification. The PCR products were concentrated, mixed 9:1, and analyzed by single-read sequencing with HiSeq 2500 (Illumina).

**Screen analysis**. Deep sequencing raw data were processed for sgRNA counting using the Python scripts. The high-throughput sequencing reads were demultiplexed using the 5 bp adapter by cutadapt[67]. The adapters of the demultiplexed reads were then trimmed by bowtie2, obtaining 20 bp gRNA sequences. These sgRNA sequences were mapped to the sequences of Human GeCKO v2 sgRNA library using bowtie2 and SAMtools[68,69]. The total number of sgRNA counts were obtained by using the MAGeCK workflow version 0.5.6, the robust rank aggregation (RRA) values and p-values were determined using the MAGeCK algorithm[21].

**Recombinant DNA**. Human GeCKOv2 CRISPR knockout pooled library (Pooled Library #1000000048)[20] and pX330-U6-Chimeric_BB-CBh-hSpCas9 (Addgene plasmid # 42230)[70] were gifts from Feng Zhang. pX330-mEGFP plasmid was previously generated from pX330-U6-Chimeric_BB-CBh-hSpCas9[71]. Human B3GALT4, UGCG, A4GALT, and B3GNT5 cDNAs were amplified by PCR from a human brain cDNA library to construct pME-hB3GALT4-3HA, pME-hUGCG-3HA, pLIB2-Hyg-hA4GALT-3HA, and pLIB2-Hyg-hB3GNT5-3HA. Human B4GALT5 were amplified by PCR from cDNA of HEK293 to construct pME-hB4GALT5-3HA. Mouse ST3GAL5 and human B4GALNT1 cDNAs were gifts from Koichi Furukawa[40,41]. Plasmid pLIB2-BSD-hB3GALT4-3HA was constructed by sub-cloning the EcoRI-NotI fragment of hB3GALT4-3HA from pME-hB3GALT4-3HA into the same sites of pLIB2-BSD. To construct pME-Puro-hB3GALT4-3FLAG, hB3GALT4 from pME-hB3GALT4-3HA digested with EcoRI and MluI was cloned into the same sites of pME-Puro-PIGX-3FLAG[66]. B3GALT4 with C-terminal FLAG-6His tags was amplified by PCR from pME-Puro-hB3GALT4-3FLAG by primer to construct pLIB2-BSD-hB3GALT4-FLAG-6His. To construct plasmids expressing 3HA tagged mutant B3GALT4, including hB3GALT4-D175A, -D176A, -D177A and -D291A, pME-hB3GALT4-3HA was used as a template and mutagenized by a commercial site-directed mutagenesis kit (Agilent Technologies). UGCG mutant plasmid, pME-hUGCG-D144A, was generated by mutagenesis of pME-hUGCG-3HA. B4GALT5 mutant plasmid, hB4GALT5-D300A, was made by mutagenesis of pME-hB4GALT5-3HA. Plasmid pLIB2-BSD-hUGCG-D144A and pLIB2-BSD-hB4GALT5-D300A were generated by sub-cloning the EcoRI-NotI fragment of pME-hUGCG-D144A and pME-hB4GALT5-D300A to the same sites of pLIB2-BSD. To make pCMV-3HA, XhoI site was introduced to pRL-CMV (Promega) by mutagenesis and digested by XhoI and NotI, then ligated with the XhoI and NotI fragment from pME-3HA. Plasmid pCMV-B3GALT4-3HA was generated by sub-cloning of PCR amplified XhoI -NotI fragment from pME-hB3GALT4-3HA into the same sites of pCMV-3HA, and the MluI site was then removed by mutagenesis. Plasmid pCMV-B4GALNT1-3HA was generated by sub-cloning of the XhoI -NotI fragment of pME-B4GALNT1-3HA to the same sites of pCMV-3HA. Plasmid pCMV-B4GALNT1-3HA was generated by sub-cloning of PCR amplified XhoI -MluI fragment from pME-PGAP4-3HA to the same sites of pCMV-3HA. Human PrP (PRNP, NM_000311) cDNA was cloned to generate plasmid pME-HA-PrP, and pME-HA-PrP-del-C was generated by using In-fusion cloning (Takara). Plasmid pME-FLAG-CD59-del-C was generated from pME-FLAG-CD59 by In-fusion cloning. Oligos for generation of the plasmids used in this study are shown in Supplementary Data 1.

**Infection of retrovirus**. PLAT-GP packaging cells were seeded at 60% confluency on 6-well plate 12 h before transfection to reach 90% confluence before transfection. Cells were transfected with pLIB2-BSD or pLIB2-Hyg plasmid bearing cDNA of interest by using PEI "Max" transfection reagent. After about 12 h culture, cells were further incubated for 12 h with 10 mM sodium butyrate. Medium was then replaced to fresh one and incubate at 32 °C for 24 h. Viral medium was added to cells and these cells were cultured at 32 °C for 16 h. On the next day, culture medium was changed and cells were cultured at 37 °C. Two days after transduction, cells were incubated in the medium with 10 µg/mL blasticidin (InvivoGen) or 400 µg/mL hygromycin B (InvivoGen) for 10 days.

**Generation of knockout cell lines**. SLC35A2, PIGO, PIGZ, B3GALT4, UGCG, B4GALT5, B4GALT6, B4GALNT1, ST3GAL5, B3GNT5, A4GALT, TMEM165, UNC50, GOSR1, STX5, UBE2J1, UBE2G2, SYVN1, DERL2, SEL1L, DERL1, OS9, XTP3B, and AMFR (gp78) knockout cells were established by CRISPR/Cas9 system. Typically, two different gRNAs targeting to exon regions of each gene were designed by CRISPOR;[72] sgRNA design for KO of B3GNT5 were previously described;[73] and pX330-mEGFP was used as the backbone for all knockout lines. PIGZ-KO, B3GALT4-KO, B4GALT5-KO, B4GALT6-KO, A4GALT-KO and UNC50-KO cell lines derived from single colonies were validated by Sanger sequencing (Supplementary Table 1). SLC35A2 KO and TMEM165 KO cells were confirmed by checking N-glycan with GS-II lectin. B4GALNT1-KO cell line was validated by checking GM1 with CTxB staining. ST3GAL5-KO cell line was confirmed by checking GM1 with CTxB and LacCer with anti-LacCer mAb. GOSR1-KO, STX5-KO, UBE2J1-KO, UBE2G2-KO and SYVN1-KO cell lines were

validated by immunoblotting to confirm the loss of each target protein (Supplementary Fig. 1g–k). KO of B3GNT5, SEL1L, DERL1, OS9, XTP3B, and AMFR were validated by PCR to confirm a large deletion of each gene.

**Quantitative real-time PCR (qRT-PCR) of B3GALT4**. Total RNA was isolated from PIGS-UGCG DKO + Vec, PIGS-UGCG DKO + UGCG, PIGS-UGCG DKO + UGCG-D144A, PIGS-B4GALT5-B4GALT6 TKO + Vec, PIGS-B4GALT5-B4GALT6 TKO + B4GALT5, PIGS-B4GALT5-B4GALT6 TKO + B4GALT5-D300A cells using the RNeasy Mini kit (QIAGEN). Each RNA was then transcribed to cDNA using the SuperScript VILO cDNA Synthesis kit (Thermo Fisher). The quantitative PCR using SYBR Green PCR Master Mix (Applied Biosystems) was performed by the StepOnePlus Real-Time System (Thermo Fisher). The RNA expression level was normalized to ACTB and the relative expression was calculated by using $-\Delta\Delta C_T$ method. The primers for qRT-PCR were listed in Supplementary Data 1.

**Microarray of PIGS-KO and PIGS-UBE2J1-DKO HEK293 cells**. Approximately $1 \times 10^6$ cells were grown in DMEM with 10% FBS overnight and harvested for RNA extraction. Total RNAs were extracted using the RNeasy Mini kit (QIAGEN) according to the manufacturer's instructions. Total RNAs were then used to generate cDNAs, and the cDNAs were biotinylated, fragmented, and hybridized on Clariom S Array, Human. Arrays was washed and stained in the GeneChip Fluidics Station 450 (Thermo Fisher Scientific). Clariom S Array was scanned using GeneChip Scanner 3000 7G System (Thermo Fisher Scientific). The data were analyzed by Affymetrix Transcriptome Analysis Console 4.1 offered SST-RMA for gene expression level analysis. Gene expression fold change higher than 2 or lower than −2 was set as default filter criteria to identify expression differences.

**Purification of HFGF-CD59 released from cells by PI-PLC**. For purification of HFGF-CD59, approximately $1 \times 10^8$ HEK293, HEK293-B3GALT4 KO and HEK293-B3GALT4-PIGZ DKO cells stably expressing HFGF-CD59 were treated with 0.5 unit/mL PI-PLC (Thermo Scientific) in 5 mL of PI-PLC buffer (Opti-MEM containing 10 mM HEPES-NaOH (pH 7.4), 1 mM EDTA and 0.1% BSA (Nacalai Tesque) at 37 °C for 2.5 h. HFGF-CD59 was also prepared from vector- or 3HA-B3GALT4-transfected HEK293-B3GALT4-PIGZ DKO cells. Supernatants were loaded on the columns filled with 0.25 mL of Glutathione Sepharose 4B (GE healthcare) at 4 °C. After washing with 5 mL PBS, HFGF-CD59 was eluted by 5 mL elution-buffer A (PBS containing 30 mM HEPES-NaOH (pH 7.4) and 20 mM reduced glutathione (Wako)). 100% cold trichloroacetic acid (Wako) was added to eluted samples to final 10 % followed by incubation on ice for 30 min. Proteins were precipitated by centrifugation at $12,000 \times g$ for 15 min at 4 °C. After twice washing with 100 % cold ethanol, pellets were resolved in $1 \times$ SDS-sample buffer with 5% β-mercaptoethanol and boiled at 60 °C for 1 h. Samples were subjected to SDS-PAGE, Coomassie brilliant blue staining (Imperial Protein Stain, Thermo Scientific), and in-gel digestion with trypsin.

**Mass spectrometry analysis**. To analyze the GPI structures derived from HFGF-CD59, the bands of purified HFGF-CD59 were excised, reduced with 10 mM dithiothreitol (DTT) and alkylated with 55 mM iodoacetamide. After in-gel digestion with trypsin, the resultant peptides were subjected to mass spectrometry analyses on LC-ESI system as previously described[5]. Briefly, LC-MS/MS analysis was performed in the positive ion mode on a nanoLC system (Advance, Michrom BioRescources) using a C18 column ($0.1 \times 150$ mm) coupled to an LTQ Qrbitrap Velos mass spectrometer (Thermo Scientific). Each sample was injected onto the column and eluted in gradients from 5 to 35% B in 45 min at 500 nL/min (Solvent A, $H_2O$; Solvent B, acetonitrile; both containing 0.1% (v/v) formic acid). The mass scanning range was set at $m/z$ 350–1500. The ion spray voltage was set at 1.8 kV. The MS/MS analysis was performed by automatic switching between MS and MS/MS modes at collision energy 35%. The obtained mass data were then analyzed by Xcalibur 2.2 (Thermo Fisher). To determine GPI species, those fragments derived from GPI anchors found in the MS/MS profiles, including fragment ions of $m/z$ $422^+$ and $447^+$, were selected, and assigned to determine the structural variations of GPI (Supplementary Fig. 3). Based on the MS/MS profiles, the peak areas of the parental MS fragments corresponding to predicted GPI containing peptides were measured and the relative amount of each GPI peptides was calculated (Supplementary Table 2).

**Inhibitors treatment**. PIGS-KO, PIGS-B3GALT4-DKO, PIGS-UGCG-DKO, and PIGS-B4GALT5-B4GALT6-TKO HEK293 cells were maintained in 5 µM myriocin or 25 µM fumonisin B1 containing DMEM medium for 3 days before FACS analysis. HeLa cells stably expressing FLAG-6His tagged human B3GALT4 were pre-treated with 50 µg/mL brefeldin A for 30 min before immunofluorescence staining.

**Flow cytometry**. Cells stained with T5 mAb, anti-DAF (CD55), anti-CD59, and anti-CD17 (LacCer) in FACS buffer (PBS containing 1% BSA and 0.1% $NaN_3$) were incubated on ice for 25 min. Cells were then washed twice in FACS buffer followed by staining with Alexa Fluor 488 or 647-conjugated goat anti-mouse IgM

for T5 mAb and anti-CD17, PE-conjugated goat anti-mouse IgG for anti-DAF or anti-CD59, and APC or PE-conjugated streptavidin for biotin-labeled anti-CD59 in FACS buffer. To analyze the glycosylation profiles of cell surfaces, cells were stained by Alexa Fluor 488-conjugated CTxB or Alexa Fluor 647-conjugated lectins in FACS buffer containing 1 mM $CaCl_2$, 1 mM $MnCl_2$, and 1 mM $MgCl_2$ on ice for 15 min. After twice washing by FACS buffer, cells were analyzed by the BD FACSCanto II.

**PI-PLC treatment.** Cells were detached from culture plates by treatment with detaching buffer, PBS containing 5 mM EDTA and 0.5% BSA. Approximately $5 \times 10^5$ cells were treated with 1 unit/ml or without PI-PLC in 50 μL reaction buffer (four volumes of Opti-MEM and one volume of detaching buffer) at 37 °C for 2 h before analysis.

**In vivo metabolic labeling of glycosphingolipids.** About $1 \times 10^6$ cells were cultured in normal medium overnight, cells were washed with Opti-MEM and pulse-labeled with 1 μCi of D-[1-$^{14}$C] Galactose (American Radiolabeled Chemicals) in 1 mL of Opti-MEM supplemented with 5% dialyzed FBS (Gibco) for 6 h. Cells were then rinsed with cold PBS, harvested by a cell scraper, pelleted, and stored at −80 ° C. The lipids were extracted by Bligh and Dyer method. Extracts were dried under nitrogen stream. Phospholipids were hydrolyzed in 0.1 M NaOH in methanol at 40 °C for 2 h. The solution was neutralized with 0.1 M HCl, 200 μL of hexane was added to wash methanol, the lower layer was evaporated and extracted by Bligh and Dyer method. To observe [$^{14}$C]-galactose-labeled GlcCer, LacCer, GM3 and Gb3 biosynthesis, lipids were separated by HPTLC 60 plate (Merck) using developing solvent: chloroform/methanol/0.25% aqueous $CaCl_2$ = 65/35/8. The radiolabeled sphingolipids on HPTLC plates were visualized by using a FLA 7000 analyzer (Fujifilm). Quantitative analysis was performed by JustTLC (SWEDAY).

**In vitro activity assay of GM1 synthase.** The membrane fractions from PIGS-KO, PIGS-B3GALT4-DKO, PIGS-UGCG-DKO + Vec, PIGS-UGCG-DKO + UGCG, PIGS-B4GALT5-B4GALT6-TKO + Vec, and PIGS-B4GALT5-B4GALT6-TKO + B4GALT5 HEK293 cells were used as the source of enzymes. About $2 \times 10^7$ cells were suspended in 800 μL of cold 25 mM HEPES buffer (pH 7.4) containing 0.25 M sucrose, 1 mM EDTA, and 1× protease inhibitor cocktail (Roche), and homogenized in a Dounce homogenizer (200 strokes, tightly fitting pestle). Nuclei were removed by centrifugation at $500 \times g$ for 5 min at 4 °C. The supernatant was recentrifuged for 60 min at $21,900 \times g$. The pellets were resuspended in 200 μL of 25 mM HEPES buffer (pH 7.4) containing 0.5% Triton X-100 and protease inhibitors using a Dounce homogenizer (50 strokes, loose fitting pestle), the total protein concentration of the supernatants after centrifugation was determined and adjusted to 5 mg/mL. For one reaction, 14.4 μg GM2 and 1 μCi (100 pmol) of UDP-[6-$^3$H] Galactose (American Radiolabeled Chemicals) in 40 μL reaction buffer (25 mM HEPES (pH 7.2), 10 mM $MnCl_2$, 0.4% Triton X-100, and 100 μM UDP-Gal) were sonicated for 1 min before reaction. The reaction was started by mixing 50 μg of proteins and kept at 37 °C for 2 h. The lipids were extracted by Bligh and Dyer method, and the gangliosides in the upper phase were dried by a centrifugal vacuum concentrator and separated by HPTLC 60 plate (Merck) using developing solvent: chloroform/methanol/0.25% aqueous $CaCl_2$ = 65/35/8. The radioactive products on HPTLC plates were visualized by using a FLA 7000 analyzer (Fujifilm). Quantitative analysis was performed by JustTLC (SWEDAY).

**In vivo metabolic labeling of GPI intermediates.** Approximately $2 \times 10^6$ of CHO-PIGS KO, HEK293-PIGS KO, PIGS-UBE2J1 DKO and PIGS-SYVN1 DKO cells were precultured in normal medium overnight, cells were washed with wash medium (glucose-free DMEM buffered with 20 mM HEPES) and incubated for 1 h at 37 °C in 1 mL of reaction medium (glucose-free DMEM buffered with 20 mM HEPES, pH 7.4, and supplemented with 10% dialyzed FBS (Gibco), 10 μg/mL tunicamycin (Wako), and 100 μg/mL glucose). [2-$^3$H] Mannose (American Radiolabeled Chemicals) was added to 25 Ci/ml, and the cells were incubated for 1 h at 37 °C in 5% $CO_2$. After labeling, the cells were washed twice with 1 ml of PBS and released from culture plates by cell scraper. Cells were pelleted and washed with 1 ml of cold PBS. Radiolabeled GPIs were extracted by 1-butanol and separated by HPTLC (Merck)[74], and visualized by a FLA 7000 analyzer (Fujifilm). Quantitative analysis was performed by JustTLC (SWEDAY).

**Immunoblotting.** Cell pellets were washed twice with ice-cold PBS prior to lysis in lysis buffer supplemented with 1% Triton X-100 and protease inhibitors (Roche) on ice, followed by centrifugation at $17,900 \times g$ for 15 min at 4 °C. Supernatants were collected, then mixed with 4 × SDS-sample buffer and boiled at 95 °C for 5 min. Samples were resolved on 10–20% SDS-PAGE gels to detect free GPI-Gal-NAc, otherwise on 7.5–15% gels, and analyzed by immunoblotting as previously described[18]. Briefly, samples were transferred to PVDF membranes, and the membranes were blocked at RT in Tris-buffered saline containing 0.1% Tween-20 (TBS-T) and 5% nonfat milk for 1 h, followed by incubation with primary antibodies at RT for 1 h. After washing three times with TBS-T buffer, the membranes were then incubated with horseradish peroxidase labeled secondary antibodies for 1 h at RT. After washing with TBS-T, blots were exposed to Amersham ECL Prime Western Blotting Detection Reagent (GE Healthcare) and imaged using

ImageQuant LAS 4000 Mini (GE Healthcare). The information of antibodies used for immunoblotting were described in the Antibodies and Reagents section above.

**Immunofluorescence.** For cell surface staining by CTxB and T5 mAb, PIGS-KO, PIGS-B3GALT4-DKO + Vec and PIGS-B3GALT4-DKO + B3GALT4 HEK293 cells were allowed to adhere to gelatin-coated coverslips overnight at 37 °C. Cells were rinsed with ice-cold PBS three times and incubated with 1 μg/mL Alexa Fluor 594-conjugated CTxB on ice for 20 min. After washing with ice-cold PBS, the cells were immediately fixed with 4% paraformaldehyde in PBS for 20 min at RT, cells were rinsed with PBS three times, blocked in blocking buffer (1% BSA and 0.1% NaN$_3$ in PBS) at RT for 1 h, and stained with T5 mAb (100× diluted in blocking buffer) for 1 h at RT. Following PBS washing steps, secondary antibody staining was performed with Alexa Fluor 488-conjugated goat anti-mouse IgM (500× diluted in blocking buffer) at RT for 1 h. Coverslips were then washed with PBS three times and mounted in ProLong Gold antifade reagent with DAPI (Thermo Fisher).

To detect 3HA tagged B3GALT4 in HEK293 cells, HEK293 cells transfected with 3HA tagged *B3GALT4* cDNA were seeded on the coverslips, fixed with 4% paraformaldehyde and permeabilized in blocking buffer with 0.1% saponin for 1 h at RT, and then incubated with 1 μg/mL rabbit anti-HA (1:500), and mouse anti-GM130 (1:500) diluted in blocking solution with 0.1% saponin and 2.5% goat serum (Gibco) for 1 h. After three washes in PBS for 30 min, coverslips were incubated with Alexa 594-conjugated goat anti-rabbit IgG (1:500) and Alexa 488-conjugated goat anti-mouse IgG (1:500) for 1 h. To detect FLAG tagged B3GALT4 in HeLa cells, cells were stained with mouse anti-FLAG (1:250) and rabbit anti-Calnexin (1:500), GM130 (1:500), or anti-B4GALT1 (1:500) followed by Alexa 594-conjugated goat anti-mouse IgG (1:500) and Alexa 488-conjugated goat anti-rabbit IgG (1:500) in blocking solution with 0.1% saponin and 2.5% goat serum. Cells stained with mouse anti-FLAG or anti-HA and sheep anti-TGN46 (1:500) were incubated with Alexa 488-conjugated donkey anti-mouse IgG (1:500) and Alexa 594-conjugated donkey anti-sheep IgG (1:500), in some cases, co-stained with rabbit anti-GM130 followed by Alexa 647-conjugated donkey anti-rabbit IgG (1:500) in blocking solution with 0.1% saponin and 5% normal donkey serum (Jackson ImmunoResearch). After washing, coverslips were mounted in ProLong Diamond anti-fade reagent (Thermo Fisher). Images were taken on an Olympus FV1000 laser scanning confocal microscope with a UPLSAPO oil lens (×100 magnification and 1.4 NA).

**Three-dimensional structured illumination microscopy.** Fixed HeLa cell samples were imaged with an Elyra SR-SIM system (Zeiss) equipped with a ×100 magnification/1.46 N.A. Plan-Apo oil immersion objective (DIC); 488-, and 561-nm OPSL lasers; and PCO edge sCMOS cameras. SI image stacks were acquired with a z-distance of 125 nm and with 15 raw SI images per plane (five phases, three angles). The SI raw datasets were computationally reconstructed with channel-specific measured optical transfer functions (OTFs) and a Wiener filter set to 0.002 using the Imaris software packages (Bitplane) to obtain a super-resolution 3D image stack with a lateral (x,y) resolution of ~120 nm and an axial (z) resolution of ~300 nm.

**Structure prediction of human B3GALT4.** The full sequence of B3GALT4 was submitted to I-TASSER for prediction of the model; and succeeded in constructing its homology model from the crystal structure of mouse Fringe, a beta-1,3-*N*-acetylglucosaminyltransferase modifying Notch receptors (PDB entry 2j0aA), a phospholipase D (PDB entry 1f0iA); Saccharomyces cerevisiae Mnn9 (PDB entry 3zf8A) and murine ppGaNTase-T1 (PDB entry 1xhbA). The overall model quality of the homology model analyzed by PROSA-Web shows a score (score = −3.12) in the range of low resolution of native conformations of this size of 378 residues. Figures were generated in Open-Source PyMOL 2.1.

**Quantification and statistical analysis.** Statistical analyses were done using GraphPad Prism7. Unpaired Student's *t*-test was used to evaluate comparisons between two individual groups, one-way ANOVA followed by Dunnett's post hoc test was used to evaluate multiple comparisons, $P$ values < 0.05 were considered as statistically significant. Heatmap was generated by the heatmap illustrator in TBtools (v0.665)[75]. Data were graphed in GraphPad Prism7.

**Reporting summary.** Further information on research design is available in the Nature Research Reporting Summary linked to this article.

## Data availability
The source data underlying Figs. 2, 3b and h, 4c, d and f–i, 5b, 6c and e, 7a, b and d, e, and Supplementary Figs. 5b and g–h, 6a, b, and 8c, d are provided as a Source Data file. Uncropped western blot images are available in Supplementary Fig. 9. Sequencing data are available in Supplementary Data 2. The microarray data are available in Supplementary Data 4 and from the GEO database under accession number GSE140855 (Microarray of HEK293-PIGS-KO and HEK293-PIGS-UBE2J1-DKO cells). The mass spectrometry proteomics data have been deposited to the ProteomeXchange Consortium via the PRIDE[76] partner repository with the dataset identifier PXD014226 (project name

Determination of GPI structure of human CD59 in HEK293 cells). All other data that support the findings of this study are available from the corresponding author upon reasonable request.

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

## Acknowledgements

We thank Dr. Jean-François Dubremetz (Montpellier University) for T5 4E10 mAb, Dr. Koichi Furukawa (Chubu University) for expression plasmids of B4GALNT1 and ST3GAL5, Dr. Yuko Tashima, Dr. Gun-Hee Lee, Dr. Yoshiki Yamaguchi, Dr. Soh Satoh, Dr. Junji Takeda, and Dr. Akihiro Harada for discussion, Yuki Uchikawa and Yuko Kabumoto for cell sorting, Yasuhiko Sato for 3D-SIM, and Keiko Kinoshita, Saori Umeshita, Yukari Onoe, Kana Miyanagi and Dr. Noriyuki Kanzawa for technical help. We also thank Edanz Group (www.edanzediting.com/ac) for editing a draft of this manuscript. This work was supported by JSPS and MEXT KAKENHI grants (JP16H04753 and 17H06422) to T.K. and a grant for Joint Research Project of the Research Institute for Microbial Diseases, Osaka University to M.F. and T.K.

## Author contributions

Y.W. and T.K. conceptualized and designed the study. With assistance from Y. Ma., Y. Mu., T.H., and M.F., Y.W. and Y.L. conducted the experiments. Y.T. and A.N. acquired and analyzed the MS data. Y.W. and T.K. wrote the manuscript with input from all authors.

## Competing interests

The authors declare no competing interests.
