## [Peer Review File · Nature Communications]

Reviewers' comments:

Reviewer #1 (Remarks to the Author):

In their manuscript entitled 'Cross-talks of glycosylphosphatidylinositol biosynthesis with glycosphingolipid biosynthesis and ER-associated degradation' Kinoshita and colleagues identify through a CRISPR screen genes involved in GPI anchor galactosylation. By this approach they find that genes encoding GSL synthetic enzymes and components of the ERAD machinery are required for efficient GPI processing. The authors focus on B3GALT4 (encoding GM1 Synthase) that they find to be absolutely required for GPI galactosylation. They show that B3GALT4 enzymatic activity is necessary for and that B3GALT4 overexpression stimulates GPI galactosylation. They also find that GSLs production up to LacCer promotes GPI galactosylation though GSL production was not absolutely required for GPI processing. Along similar lines the authors show that defective ERAD leads to hyperproduction of GPIs whose load exceeds the cell processing capacity thus resulting in the accumulation of inappropriately processed GPIs. This is a very interesting study where the data presented will be of interest for a broad scientific audience. Nonetheless I find it a bit unfocused whereby the part in ERAD is not developed fully and conceptually disconnected from the rest of the manuscript. I would suggest that the authors keep the ERAD part for a future and more mature communication. I also have some concerns about the GSL part that the authors need to address before resubmission.

1. The authors propose that GM1 Synthase directly acts on GPI to galactosylate it. While supported by convergent indirect evidence, this is not formally proven. Authors should provide in vitro evidence for B3GALT4 acting GPI by the use of recombinant GM1S and pure GPI.
2. The authors have addressed the involvement of different GSL series (i.e. ganglio/ asialo) to GPI processing. It would be desirable to extend this analysis to the other 2 main GSL classes (i.e., globo and lacto) by interfering with the activity of the relevant enzymes (i.e., A4GALT/Gb3S and B3GNT5/LC3S). Specifically, HeK293 are high Gb3 producers thus testing the role of this GSL in GPI processing is of relevance.
3. The authors maintain that the finding that GSL and GPI synthetic pathways crosstalk can suggest some degree of coevolution. This is an interesting idea the authors should articulate more on. Specifically, GSL synthetic system is profoundly different in fungi/plants, insects and vertebrates how did the GPI structure changes along this phylogenic axis?

Reviewer #2 (Remarks to the Author):

This is a very interesting and significant paper that shows that mammalian GPI anchor side-chain modification shares the GSL modifying GT B3GALT4 to make the Galb1-3GalNAc side-chain. Quite unexpected! Further, they demonstrate a fascinating cross talk between GSL and GPI biosynthesis in that LacCer greatly increases the efficiency of GPI galactosylation via B3GALT4 – lastly they show an equally interesting and unexpected (though logical) link between GPI biosynthesis and ERAD - such that GPI biosynthesis is downregulated in times of ER stress. These related but unexpected events and phenomena stem from a very elegant and intelligent genetic screen performed by the group . I have looked closely at the methodology and data and agree with the conclusions. The group should be congratulated on rather thoroughly investigating alternative explanations- for example, whether LacCer affects the level and/or location of B3GALT4 rather than enhancing the transfer of Gal to GPIs in situ.

Michael Ferguson

Reviewer #3 (Remarks to the Author):

In this manuscript, Wang et al identified several cellular factors involved in GPI side-chain modifications using a genome-wide CRISPR knockout library. The authors reveal some novel findings from the screening as follows:

1. B3GALT4, also called GM1 synthase, is the responsible enzyme that transfers galactose to the GalNAc side chain of GPI.
2. GSL biosynthesis is required for efficient galactosylation of the GPI side chain (GPI-GalNAc) by B3GALT4.
3. Defects in the ERAD pathway cause accumulation of GPIs when the transfer of GPI to proteins is inhibited.

Overall, the study has been well-performed with high novelty although biological significance of GPI side chain has not yet been clarified; therefore, the manuscript is worth publishing in this journal. However, proofs leading to their conclusions are a little insufficient, therefore the authors should address the following issues for publication.

Major points:

1. The authors argue that LacCer enhances B3GalT4 activity toward GPI-GalNAc. However, it is still unclear whether endogenous B3GalT4 activity is suppressed in B4GalT5-B4GalT6 KO cells.

(a) Enzymatic activity of B3GalT4 toward GM2 and GPI-GalNAc (or GPI-GalNAc-APs) should be measured in vitro to compare PIGS-B4GalT5-B4GalT6 TKO cells with PIGS single KO (parent) cells.

(b) In Fig.6, it is difficult to interpret why the effect of transient expression of B3GalT4 was different from that of stable expression. And the authors drew a conclusion from the results using transient expression although endogenous B3GalT4 is stably expressed. Please describe possible reasons.

2. It is still unclear whether LacCer itself is specifically required to enhance B3GalT4 activity or the total amount of LacCer and the subsequent GSLs affects the activity.

(a) Relative amounts of GSLs including LacCer should be measured in GSL-KO cells.

(b) LacCer is increased in ST3Gal5 KO cells as the authors demonstrated in Fig.4f. If LacCer in the PIGS-B4GalT5 DKO cells is increased by knockout of ST3Gal5 (PIGS-B4GalT5-ST3Gal5 TKO), is the accumulated GPI-GalNAc (anti T5 mAb staining) reduced through B3GalT4 activation? Please demonstrate the correlation between expression level of LacCer and B3GalT4 activity.

3. It is unclear what triggers GPI upregulation in ERAD-defective PIGS-KO cells; therefore, the authors need to provide some clues to clarify the mechanism.

(a) In Fig.7, is the transcriptional level of GPI-synthesis genes upregulated in ERAD-defective PIGS-KO cells?

(b) Does overexpression of B3GalT4 decrease GPI-GalNAc in ERAD-defective PIGS-KO cells?

(c) Does overexpression of CD59 and Prp or their mutants with deletion of GPI signal sequence up-regulate GPIs?

(d) Can the ERAD pathway be used to degrade free GPIs? Please describe the possibility.

(e) Is there any situation where transfer of GPI to proteins is suppressed by any stress? If so, please add the description to make the found phenomenon more significant biologically.

Minor points:

1. The binding amount of anti-T5 in PIGS-B4GalT5-B4GalT6 TKO cells appears to be different between in Fig.4e (Vec) and in Fig.4h (Blue line). Why is it so different?

2. In Fig.4g, “Acyl-CoA” between “Ser” and “3K5a” should be Palmitoyl CoA (Pal-CoA), and Acyl-CoA can be added between “Sphinganine” and “DHCer”.

3. Student’s t-test is not a suitable method when a sample is repeatedly used in statistical analysis, so correction for multiple testing is required.

Reviewer #4 (Remarks to the Author):

minor changes required:

Ln1/2 The title could be made more specific, highlighting the key finding(s); the connection to ERAD is evident in the data presented but the role is unclear; ERAD could therefore be left out

Ln23/24 “also called GM1 ganglioside synthase”should better be replaced by “the previously characterized GM1 ganglioside synthase” or something similar, to make sense of the word “additionally” that follows and that might otherwise be misleading in this context

Ln27 given the way the abstract/summary is currently written, the implied “evolutionary relationship” is not obvious; in the discussion section the relationship between GPI and GSL pathways are discussed but that is not evident in the abstract/summary

Ln179/Ln180 a connection of GPI in the context of GPI-APs and ERAD has been reported previously:

Fujita, M., O.T. Yoko, and Y. Jigami. 2006. Inositol deacylation by Bst1p is required for the quality control of glycosylphosphatidylinositol-anchored proteins. *Mol Biol Cell*. 17:834–850.

Sikorska, N., L. Lemus, A. Aguilera-Romero, J. Manzano-Lopez, H. Riezman, M. Muniz, and V. Goder. 2016. Limited ER quality control for GPI-anchored proteins. *J Cell Biol*. 213:693–704.

Ln488 “E2 ligases” is misleading, use E2 enzymes instead

Fig7g is currently unclear; in my opinion, it does not display an easily comprehensible model

Ln561-64 A physical interaction of LacCer and GPI-APs is not shown in the paper, however, the authors show a functional dependency of GPI anchor modification with Gal on LacCer; that is not incompatible with the model that GPI-APs and GSLs interact physically only at the PM

Ln608-617 It has been a long-standing question in the field how free GPIs can leave the ER and how that would relate to the export of GPI in the context of GPI-APs. For instance, GPI remodeling is a prerequisite for efficient ER export and that in turn depends to a large degree on attachment to a protein (substrate). Can the authors comment on this?

The fact that ERAD seems to be able to regulate GPI abundance only in the absence of the GPI-Transferase is difficult to understand from physiologically point of view, because the transferase activity is essential. The relevance of this observation is still unclear, can the authors provide a more plausible model for future investigation?

Reviewer #1 (Remarks to the Author):

In their manuscript entitled ‘Cross-talks of glycosylphosphatidylinositol biosynthesis with glycosphingolipid biosynthesis and ER-associated degradation’ Kinoshita and colleagues identify through a CRISPR screen genes involved in GPI anchor galactosylation. By this approach they find that that genes encoding GSL synthetic enzymes and components of the ERAD machinery are required for efficient GPI processing. The authors focus on B3GALT4 (encoding GMI Synthase) that they find to be absolutely required for GPI galactosylation. They show that B3GALT4 enzymatic activity is necessary for and that B3GALT4 overexpression stimulates GPI galactosylation. They also find that GSLs production up to LacCer promotes GPI galactosylation though GSL production was not absolutely required for GPI processing. Along similar lines the authors show that defective ERAD leads to hyperproduction of GPIs whose load exceeds the cell processing capacity thus resulting in the accumulation of inappropriately processed GPIs. This is a very interesting study where the data presented will be of interest for a broad scientific audience.

We thank the reviewer for the support and thoughtful suggestions. We addressed points raised by the reviewer as much as possible and made revisions accordingly.

Nonetheless I find it a bit unfocussed whereby the part in ERAD is not developed fully and conceptually disconnected form the rest of the manuscript. I would suggest that the authors keep the ERAD part for a future and more mature communication.

We agree to the reviewer that the ERAD part is still incomplete. We are continuing study to clarify the molecular mechanisms of GPI upregulation. A finding of involvement of ERAD in GPI biosynthesis regulation is new and a report of the initial finding at this time will be useful for readers (We added some more data according to reviewer #3's comments). We do hope that the reviewer agrees to this view.

I also have some concerns about the GSL part that the authors need to address before resubmission.

1. The authors propose that GM1 Synthase directly acts on GPI to galactosylate it. While supported by convergent indirect evidence, this is not formally proven. Authors should provide in vitro evidence for B3GALT4 acting GPI by the use of recombinant GM1S and pure GPI.

We agree to the reviewer's comment that in vitro study with pure components is the best way to demonstrate B3GALT4 activity towards GPI-GalNAc. However, pure GPI-GalNAc was not readily available because chemical synthesis is demanding. Instead, as an acceptor substrate we prepared GPI-AP 6His-FLAG-GST-FLAG-tagged CD59 bearing GalNAc from HEK293-B3GALT4-KO cells, quality of which was confirmed by both silver staining and Western blotting (**b** in Figure shown below). We also purified FLAG-6His-tagged B3GALT4 (GM1 synthase) expressed in HEK293 cells, which was confirmed by both Imperial Protein Stain and Western blotting (**a**). Using UDP-galactose and GM2 as the known acceptor substrate for B3GALT4, we detected a strong activity by measuring released UDP by UDP-Glo Glycosyltransferase Assay, confirming that purified B3GALT4 was active (left part of **c**). Then we tried the same B3GALT4 for the tagged CD59 bearing GalNAc with or without LacCer in the presence of UDP-galactose. However, only a tiny amount of UDP was generated in the presence of LacCer and we were not sure whether it is significant (right part of **c**). The lack of clear activity might be due to the very low concentration of CD59 bearing GPI-GalNAc or unsuitable reaction conditions or both. Although we are not able to provide a formal demonstration of galactosyltransferase activity of B3GALT4 toward GPI-GalNAc at this stage, we believe that all the supporting data together provide a convincing evidence that B3GALT4 galactosylates GPI and warrant publication of this conclusion.

2. The authors have addressed the involvement of different GSL series (i.e. ganglio/asialo) to GPI processing. It would be desirable to extend this analysis to the other 2 main GSL classes (i.e., globo and lacto) by interfering with the activity of the relevant enzymes (i.e., A4GALT/Gb3S and B3GNT5/LC3S). Specifically, HeK293 are high Gb3 producers thus testing the role of this GSL in GPI processing is of relevance.

Thanks for this suggestion. We did this experiment to test whether globo and lacto GSLs are required for GPI galactosylation. KO of A4GALT or B3GNT5 in PIGS-KO cells did not significantly affect the galactosylation of GPI-GalNAc, further supporting the conclusion that LacCer is required for galactosylation of GPI in cells. We included the results in new Fig. 4g.

3. The authors maintain that the finding that GSL and GPI synthetic pathways crosstalk can suggest some degree of coevolution. This is an interesting idea the authors should articulate more on. Specifically, GSL synthetic system is profoundly different in fungi/plants, insects and vertebrates how did the GPI structure changes along this phylogenic axis?

Thank you for this suggestion. We used “evolutionary relationship” to mean that GPI and GSL pathways share the same enzyme B3GALT4 in mammalian cells. Because of the word number limitation of the Abstract, we could not add more. To avoid any

possible misleading, we removed evolutionary and kept only “functional relationships”. Information on GPI side-chain structures in various organisms is very limited and not sufficient to discuss coevolutionary relationships with GSL structures. Our point regarding relationships between GPI and GSL applies only to mammalians at the moment.

Reviewer #2 (Remarks to the Author):

This is a very interesting and significant paper that shows that mammalian GPI anchor side-chain modification shares the GSL modifying GT B3GALT4 to make the Galb1-3GalNAc side-chain. Quite unexpected! Further, they demonstrate a fascinating cross talk between GSL and GPI biosynthesis in that LacCer greatly increases the efficiency of GPI galactosylation via B3GALT4 – lastly they show an equally interesting and unexpected (though logical) link between GPI biosynthesis and ERAD - such that GPI biosynthesis is downregulated in times of ER stress. These related but unexpected events and phenomena stem from a very elegant and intelligent genetic screen performed by the group . I have looked closely at the methodology and data and agree with the conclusions. The group should be congratulated on rather thoroughly investigating alternative explanations- for example, whether LacCer affects the level and/or location of B3GALT4 rather than enhancing the transfer of Gal to GPIs in situ.

Michael Ferguson

Thank you very much for the highly positive comments to this work.

Reviewer #3 (Remarks to the Author):

In this manuscript, Wang et al identified several cellular factors involved in GPI side-chain modifications using a genome-wide CRISPR knockout library. The authors reveal some novel findings from the screening as follows:

1. *B3GALT4, also called GM1 synthase, is the responsible enzyme that transfers galactose to the GalNAc side chain of GPI.*
2. *GSL biosynthesis is required for efficient galactosylation of the GPI side chain (GPI-GalNAc) by B3GALT4.*
3. *Defects in the ERAD pathway cause accumulation of GPIs when the transfer of GPI to proteins is inhibited.*

Overall, the study has been well-performed with high novelty although biological significance of GPI side chain has not yet been clarified; therefore, the manuscript is worth publishing in this journal. However, proofs leading to their conclusions are a little insufficient, therefore the authors should address the following issues for publication.

We thank the reviewer for the constructive suggestions. We addressed the issues raised by the reviewer and included them into the revised manuscript.

Major points:

1. The authors argue that LacCer enhances B3GalT4 activity toward GPI-GalNAc. However, it is still unclear whether endogenous B3GalT4 activity is suppressed in B4GalT5-B4GalT6 KO cells.

(a) Enzymatic activity of B3GalT4 toward GM2 and GPI-GalNAc (or GPI-GalNAc-APs) should be measured in vitro to compare PIGS-B4GalT5-B4GalT6 TKO cells with PIGS single KO (parent) cells.

Thank you for this suggestion. To determine endogenous B3GALT4 enzymatic activity in LacCer-deficient cells, we incubated cell lysates with GM2 and UDP-[6-³H]galactose, and determined generation of radioactive GM1. GM1 synthase activities of B3GALT4 in PIGS-UGCG-DKO and PIGS-B4GALT5-B4GALT6-TKO cells were only slightly lower than parent PIGS-KO cells (new Supplementary Fig. 6b). Therefore, it is not likely that the nearly complete lack of GPI galactosylation in LacCer-deficient cells is accounted for by a loss of B3GALT4 enzyme. As explained in the answer to reviewer #1, in vitro enzyme activity of B3GALT4 for GPI galactosylation cannot be measured in our hands at this moment.

(b) In Fig.6, it is difficult to interpret why the effect of transient expression of B3GalT4 was different from that of stable expression. And the authors drew a conclusion from the results using transient expression although endogenous B3GalT4 is stably expressed. Please describe possible reasons.

Situations in experiments with transient and stable expression of B3GALT4 are different. Before transfection, LacCer-deficient cells had GPIs without galactose, therefore, were positively stained by T5 mAb. To see the reduction of T5 staining, preexisting GPIs without galactose need to turn over, which requires several cell divisions. In transient transfection system, T5 staining was done in a few days after transfection and the reduction of the preexisting T5 positive staining was insufficient. In contrast, T5 staining was done a week after transfection in stable transfection system, therefore, there was no “background” T5 staining. This must be the reason for the difference between transient and stable transfection systems. To compare efficiencies of GPI galactosylation in the presence and absence of LacCer based on reduction of T5 staining, we needed transient system because stable transfection system gave only all or none results. We included these points in the revised manuscript in page 20, lines 481-487.

2. It is still unclear whether LacCer itself is specifically required to enhance B3GalT4 activity or the total amount of LacCer and the subsequent GSLs affects the activity.

(a) Relative amounts of GSLs including LacCer should be measured in GSL-KO cells.

We cultured various mutant cells with [¹⁴C]-galactose for 6 hours to determine GSL profiles. Cells defective in galactosylation of GPI lacked LacCer (new Figure 4h and Supplementary Figure 5e). Therefore, GPI-galactosylation by B3GALT4 was correlated with LacCer levels. Since the generation of the subsequent GSLs depends on LacCer, this result did not suggest that only LacCer itself modulates GPI galactosylation in vivo. The more complex GSLs might also be able to enhance B3GALT4 activity, although LacCer is the minimum GSL which enhances B3GALT4 activity to GPI-GalNAc. We included this in the revised manuscript (page 14, lines 353-358).

(b) LacCer is increased in ST3Gal5 KO cells as the authors demonstrated in Fig.4f. If LacCer in the PIGS-B4GalT5 DKO cells is increased by knockout of ST3Gal5 (PIGS-B4GalT5-ST3Gal5 TKO), is the accumulated GPI-GalNAc (anti T5 mAb staining) reduced through B3GalT4 activation? Please demonstrate the correlation between expression level of LacCer and B3GalT4 activity.

Thank you very much for this good idea. As suggested, we generated PIGS-B4GalT5-ST3Gal5 TKO cells and determined levels of T5 mAb staining. Compared to PIGS-B4GalT5 DKO cells, T5 mAb staining intensity clearly decreased by ST3Gal5 KO (We added the results as new Figure 4i), which should due to an increase of LacCer in the Golgi, although LacCer was not observed on the cell surfaces of PIGS-B4GALT5-ST3GAL5-TKO cells. This result further supports a conclusion that LacCer itself is functional in the Golgi to enhance B3GALT4 activity towards GPI-GalNAc, and suggests that more complex GSLs generated from LacCer, such as GM3, might be quickly transported to the plasma membrane. Taken together, the correlation between LacCer itself and B3GALT4 activity towards GPI is demonstrated.

3. It is unclear what triggers GPI upregulation in ERAD-defective PIGS-KO cells; therefore, the authors need to provide some clues to clarify the mechanism.

(a) In Fig.7, is the transcriptional level of GPI-synthesis genes upregulated in ERAD-defective PIGS-KO cells?

We plan to investigate molecular mechanisms leading to upregulation of GPI synthesis when ERAD is defective in the next stage of the study. As pointed out by the reviewer, because regulation at the transcriptional level of GPI-synthesis genes is a possible mechanism of GPI synthesis upregulation and because it takes relatively a short time period to address experimentally, we compared transcript levels in PIGS-KO cells and PIGS-UBE2J1 DKO cells by microarray analysis. Transcript levels of all PIG and PGAP genes were not different (changes within +/-2) (data is added as new Fig. 7h, Supplementary Fig. 7b and Supplementary Table 5). Therefore, this mechanism is unlikely to be true (described in page 23, lines 565-571). Another possible mechanism is regulation at the GPI biosynthetic enzymes such as stabilization of enzyme proteins

or enhancement of enzyme activity in ERAD-defective cells. We will ask these mechanisms in future studies.

(b) Does overexpression of B3GalT4 decrease GPI-GalNAc in ERAD-defective PIGS-KO cells?

Thank you for the suggestion. We did this experiment and found that transfection of B3GALT4 cDNA significantly decreased T5 mAb staining intensities of PIGS- and ERAD-defective cells. These results documented that the positive T5 mAb staining caused by ERAD disruption was in fact due to insufficient galactosylation capacity of the cells. These data are now shown as new Figure 7d and described in Results (p23, lines 546-549). Because of this addition, former Figure 7d, e, and f are Figure 7e, f, and g, respectively in the revised manuscript.

(c) Does overexpression of CD59 and Prp or their mutants with deletion of GPI signal sequence up-regulate GPIs?

Thank you for the idea. We did the suggested experiment to test the idea that precursor proteins of GPI-APs might cause upregulation of GPI synthesis. As shown in new Supplementary Fig. 8a-d, transfection of FLAG-tagged CD59 cDNA or HA-tagged PrP cDNA into PIGS-KO cells did not induce positive T5 mAb staining. Transfection of these cDNAs into PIGS-UBE2J1-DKO cells did not increase levels of T5 staining. Their mutant cDNAs lacking GPI signal were not effective either. FLAG-tagged CD59 cDNA did not generate detectable CD59 protein in PIGS-KO cells, whereas transfection of the same cDNA into PIGS- UBE2J1-DKO cells generated clearly detectable level of CD59 protein (Supplementary Fig. 8b). The mutant cDNA lacking GPI signal generated CD59 in both cell types. These results, therefore, indicate that CD59 with GPI signal is efficiently degraded by ERAD and does not accumulate in PIGS-KO cells. Therefore, it was not clear whether accumulation of CD59 bearing GPI signal causes upregulation of GPI. In contrast, HA-tagged PrP bearing GPI signal was overexpressed and accumulated at similar levels in PIGS-KO cells and PIGS- UBE2J1-DKO cells. So, accumulation of PrP bearing GPI signal did not cause upregulation of GPI. However, it is still possible that precursors of many GPI-APs together function as the trigger of

enhancement of GPI biosynthesis. We described these results in page 24, lines 573-594. We plan to further investigate the mechanism in the next study.

(d) Can the ERAD pathway be used to degrade free GPIs? Please describe the possibility.

Our results indicated that increase of free GPIs in the absence of ERAD is caused by upregulation of biosynthesis (Figure 7e). Therefore, it seems unlikely that ERAD is involved in degradation of free GPIs. We added this point in Discussion (page 28, lines 685-687).

(e) Is there any situation where transfer of GPI to proteins is suppressed by any stress? If so, please add the description to make the found phenomenon more significant biologically.

More than 50 patients with inherited GPI deficiency caused by partial loss of GPI-transamidase have been reported. In these patients' cells, GPI might be excessive to the transfer capacity of the transamidase, leading to accumulation of GPI if not properly regulated. Other possible situation to have excessive GPI might occur with ER stress where translation of secretory proteins including GPI-AP precursors is generally downregulated. Down regulation of GPI synthesis might also be required to prevent GPI accumulation. We added these points in Discussion (page 28, lines 690-696 in Discussion).

Minor points:

1. The binding amount of anti-T5 in PIGS-B4GalT5-B4GalT6 TKO cells appears to be different between in Fig.4e (Vec) and in Fig.4h (Blue line). Why is it so different?

We think it might be caused by compensation of some lipids after longer time culturing of these cells. We repeated analysis for PIGS-B4GALT5-B4GALT6-TKO cells and included the new data (now in Supplementary Fig. 5h).

2. In Fig.4g, “Acyl-CoA” between “Ser” and “3K5a” should be Palmitoyl CoA (Pal-CoA), and Acyl-CoA can be added between “Sphinganine” and “DHCer”.

We edited Fig. 4g as suggested (now Supplementary Fig. 5f). Thank you for the suggestion.

3. Student’s t-test is not a suitable method when a sample is repeatedly used in statistical analysis, so correction for multiple testing is required.

Thank you for the instruction. We used ANOVA followed by Dunnett’s multiple comparisons test to replace previous student’s t-test in Fig. 3h, Fig. 6c, and Supplementary Fig. 5g and h. Previous Student’s t-test in Fig. 2c and Fig. 7a were just kept, because the comparisons are complementary and the results are consistent, we think there is no need to correct for multiple comparisons for them (Rothman, K.J. (1990). No adjustments are needed for multiple comparisons. *Epidemiology*, 1: 43-46).

Reviewer #4 (Remarks to the Author):

minor changes required:

Ln1/2 The title could be made more specific, highlighting the key finding(s); the connection to ERAD is evident in the data presented but the role is unclear; ERAD could therefore be left out

We intentionally made a title without having specific findings for this Journal with a wide readership. We thought that because the title is immediately followed by the Abstract which includes specific key findings, a general title would better attract interests of potential readers.

Ln23/24 “also called GM1 ganglioside synthase”should better be replaced by “the previously characterized GM1 ganglioside synthase” or something similar, to make

sense of the word “additionally” that follows and that might otherwise be misleading in this context

We edited this as suggested. Thank you for the suggestion.

Ln27 given the way the abstract/summary is currently written, the implied “evolutionary relationship” is not obvious; in the discussion section the relationship between GPI and GSL pathways are discussed but that is not evident in the abstract/summary

We used “evolutionary relationship” to mean that GPI and GSL pathways share the same enzyme B3GALT4. Because of the word number limitation of the Abstract, we could not add more. To avoid any possible misleading, we removed evolutionary and kept only “functional relationships”.

Ln179/Ln180 a connection of GPI in the context of GPI-APs and ERAD has been reported previously:

Fujita, M., O.T. Yoko, and Y. Jigami. 2006. Inositol deacylation by Bst1p is required for the quality control of glycosylphosphatidylinositol-anchored proteins. Mol Biol Cell. 17:834–850.

Sikorska, N., L. Lemus, A. Aguilera-Romero, J. Manzano-Lopez, H. Riezman, M. Muniz, and V. Goder. 2016. Limited ER quality control for GPI-anchored proteins. J Cell Biol. 213:693–704.

Thank you for pointing to this. Citing two references, we revised the sentence to indicate that whereas ERAD was implicated in degradation of misfolded GPI-APs, its relationship to GPI has not been reported (page 7, lines 178-179).

Ln488 “E2 ligases” is misleading, use E2 enzymes instead

We fixed it. Thank you.

Fig7g is currently unclear; in my opinion, it does not display an easily comprehensible

model

We modified it and provided a new model based on our current data (now in Fig. 7i).

Ln561-64 A physical interaction of LacCer and GPI-APs is not shown in the paper, however, the authors show a functional dependency of GPI anchor modification with Gal on LacCer; that is not incompatible with the model that GPI-APs and GSLs interact physically only at the PM

We did not show physical interaction of LacCer with GPI-APs in the Golgi, and it is possible that LacCer-dependent action of B3GALT4 to GPI-APs does not involve physical interaction of LacCer with GPI-APs. To avoid misunderstanding that we suggest physical interaction between LacCer and GPI-APs, we added “functional” to clearly indicate what we demonstrate is “functional interaction” but not physical interaction (Discussion, page 26, line 373).

Ln608-617 It has been a long-standing question in the field how free GPIs can leave the ER and how that would relate to the export of GPI in the context of GPI-APs. For instance, GPI remodeling is a prerequisite for efficient ER export and that in turn depends to a large degree on attachment to a protein (substrate). Can the authors comment on this?

We previously studied remodeling of free GPIs during transport from the ER to the cell surface (Wang Y et al, J Biol Chem, 2019). Our data showed that free GPIs undergo similar remodeling reactions to GPI-APs in the ER. Therefore, structure of free GPIs become competent for binding to p24 cargo receptor complex. We further showed free GPIs undergo fatty acid remodeling. So, free GPIs are transported to the cell surface through the Golgi, perhaps excluding a possibility of contact site dependent ER-to-PM transport.

The fact that ERAD seems to be able to regulate GPI abundance only in the absence of the GPI-Transferase is difficult to understand from physiologically point of view,

because the transferase activity is essential. The relevance of this observation is still unclear, can the authors provide a more plausible model for future investigation?

More than 50 patients with inherited GPI deficiency caused by partial loss of GPI-transamidase have been reported. In these patients' cells, GPI might be excessive to the transfer capacity of the transamidase, leading to accumulation of GPI if not properly regulated. Other possible situation to have excessive GPI might occur with ER stress where translation of secretory proteins including GPI-AP precursors is generally downregulated. Down regulation of GPI synthesis might also be required to prevent GPI accumulation. We added these points in Discussion (p28, last paragraph in Discussion).

REVIEWERS' COMMENTS:

Reviewer #1 (Remarks to the Author):

The authors have fully addressed my concerns. I therefore recommend this manuscript for publication

Reviewer #3 (Remarks to the Author):

In the revised version, the authors added new data and text to answer the reviewers' comments. One of the points that should be made clear is whether LacCer itself specifically enhance B3GalT4 activity, and the authors demonstrated the correlation between expression level of LacCer and B3GalT4 activity. Other revised parts are also reasonable to answer the comments, so the paper is now thought to be acceptable for the publication in this journal.